# Potent antitumour activity of interleukin-2-Fc fusion proteins requires Fc-mediated depletion of regulatory T-cells

Rodrigo Vazquez-Lombardi[1,2], Claudia Loetsch[1,2], Daniela Zinkl[1], Jennifer Jackson[1], Peter Schofield[1], Elissa K. Deenick[1,2], Cecile King[1,2], Tri Giang Phan[1,2], Kylie E. Webster[1,2], Jonathan Sprent[1,2] & Daniel Christ[1,2]

Interleukin-2 (IL-2) is an established therapeutic agent used for cancer immunotherapy. Since treatment efficacy is mediated by CD8$^+$ and NK cell activity at the tumour site, considerable efforts have focused on generating variants that expand these subsets systemically, as exemplified by IL-2/antibody complexes and 'superkines'. Here we describe a novel determinant of antitumour activity using fusion proteins consisting of IL-2 and the antibody fragment crystallizable (Fc) region. Generation of long-lived IL-2-Fc variants in which CD25 binding is abolished through mutation effectively prevents unwanted activation of CD25$^+$ regulatory T-cells (Tregs) and results in strong expansion of CD25$^-$ cytotoxic subsets. Surprisingly, however, such variants are less effective than wild-type IL-2-Fc in mediating tumour rejection. Instead, we report that efficacy is crucially dependent on depletion of Tregs through Fc-mediated immune effector functions. Our results underpin an unexpected mechanism of action and provide important guidance for the development of next generation IL-2 therapeutics.

[1] Immunology Division, Garvan Institute of Medical Research, Sydney, New South Wales 2010, Australia. [2] St Vincent's Clinical School, University of New South Wales, Sydney, New South Wales 2010, Australia. Correspondence and requests for materials should be addressed to J.S. (email: j.sprent@garvan.org.au) or to D.C. (email: d.christ@garvan.org.au).

nterleukin-2 (IL-2) is a pleiotropic cytokine essential for the development, activation and homoeostasis of multiple lymphocyte subsets[1]. Initially identified as a potent T-cell proliferating factor present in mixed leukocyte cultures[2], IL-2 was first cloned and synthesized in *Escherichia coli* in 1983 (refs 3,4) and underwent initial clinical evaluation for cancer indications in 1985 (ref. 5). Despite severe toxicity, the potent antitumour activity observed in a subset of patients led to the regulatory approval of a high-dose recombinant IL-2 formulation for cancer immunotherapy of metastatic renal cancer in 1992 and for metastatic melanoma in 1998 (ref. 6).

IL-2 is a member of the common gamma-chain ($\gamma_c$) cytokine family and shares the $\gamma_c$ receptor subunit with IL-4, IL-7, IL-9, IL-15 and IL-21. Importantly, immune cells express dimeric or trimeric IL-2 receptors (IL-2R), with the former composed of IL-2R$\beta$ (CD122) and $\gamma_c$ (IL-2R$\beta\gamma_c$ intermediate-affinity receptor, $K_D \sim 1$ nM), and the latter composed of IL-2R$\alpha$ (CD25), IL-2R$\beta$ and $\gamma_c$ (IL-2R$\alpha\beta\gamma_c$ high-affinity receptor, $K_D \sim 10$ pM)[7,8]. Dimeric receptors are expressed on cytotoxic CD8$^+$ T-cells and natural killer (NK) cells, while trimeric receptors are predominantly displayed on activated lymphocytes and CD4$^+$ CD25$^+$ FoxP3$^+$ regulatory T-cells (Tregs)[1]. Due to its increased affinity for trimeric receptors, IL-2 induces preferential stimulation of Tregs, which are crucial for maintaining immune tolerance and display pro-tumourigenic activity[9]. In addition to the undesired promotion of Treg proliferation, clinical use of IL-2 is further complicated by a short serum half-life ($\sim 7$ min) and dose-limiting toxicities[10,11].

These shortcomings are greatly improved when IL-2 is complexed with anti-IL-2 antibodies. IL-2/mAb complexes display a substantial prolongation of serum half-life and modulate IL-2 selectivity for specific immune cell subsets[12]. In particular, IL-2/mAb complexes that target IL-2 to cells expressing CD122, but not CD25 (such as Tregs), induce preferential expansion of CD122$^{high}$ populations (that is, memory-phenotype (MP) CD8$^+$ T-cells and NK cells), leading to increased antitumour activity at low doses and reduced toxicity in animal models of malignancy[13]. The favourable properties of IL-2/mAb complexes result from a combination of factors including reduced renal clearance, FcRn recycling, and steric blockade of the CD25-binding site[14]. In addition to IL-2 immunocomplexing, modulation of activity has been reported for engineered IL-2 'superkine' variants with altered binding to components of the IL-2 receptor, namely binding to IL-2R$\alpha$ (refs 15–17), IL-2R$\beta$ (refs 16,18,19) and $\gamma_c$ (refs 16,19).

Although cytokine–antibody complexes and variants are considerably more potent than unmodified cytokines[12,18,20,21], development into validated human therapeutics has so far not been demonstrated. While this is likely a reflection of their relative recent discovery, the need for humanization of the antibody component and the requirement for formulating multiple proteins complicate development and regulatory approval of complexes, whereas IL-2 superkines suffer from short serum half-lives, due to their low molecular mass and absence of half-life extension. By contrast, Fc-fusion proteins generated through the genetic linkage of antibody Fc regions with an effector moiety (such as a cytokine or cytokine receptor) have a well-established track record as human therapeutics, as exemplified by the TNFR2-Fc-fusion protein etanercept (Enbrel)[22]. Indeed, a large proportion of new biologic drugs contain antibody Fc regions due to the commercial requirement for half-life extension[23], further highlighting the relevance of this format for drug development applications. As such, a number of IL-2-Fc-fusion proteins with therapeutic potential for induction of transplantation tolerance[19,24–26] and prevention of autoimmunity[27,28] have been reported. Furthermore, recent studies highlight the synergistic nature of combination therapy strategies consisting of antitumour antigen antibodies and IL-2-IgG fusions in models of malignancy[29,30]. Here we apply the Fc-fusion protein concept to the IL-2 system and systematically investigate the contribution of cytokine and Fc components to antitumour activity.

## Results

**Abolition of CD25 binding enhances IL-2-Fc selectivity.** We first generated genetic fusions, by linking human IL-2 with the Fc region of murine IgG2c by means of a short glycine-serine linker (see Methods). The fusion proteins display a molecular weight of $\sim 80$ kDa (Supplementary Fig. 1A) compared to about 15 kDa for human IL-2. As observed for IL-2/mAb complexes, the presence of the Fc component results in a molecular mass well above the glomerular filtration cut-off ($\sim 60$ kDa), and increased serum half-life through reduced renal clearance and FcRn recycling[31].

To reduce binding of our IL-2-Fc construct to CD25$^+$ cells (and Tregs in particular), we introduced mutations directed at disrupting the IL-2/CD25 interaction (Fig. 1a, Supplementary Fig. 1). Inspection of the quaternary IL-2/IL-2R complex structure[32] revealed that many of the residues in the IL-2/CD25 interface are charged and participate in electrostatic interactions with the receptor. This motivated us to introduce charge-reversal mutations at contact positions in order to considerably reduce binding affinity and activation of CD25$^+$ cells (see Supplementary Discussion). Our strategy relied on the introduction of charge-reversal substitutions directly into bivalent IL-2-Fc constructs, thus differing from previous approaches that have targeted both charged and aromatic residues in monovalent unfused IL-2 (refs 15,33).

Although binding was considerably reduced, residual affinity to CD25 was observed for all designed single mutations, as well as for a previously reported F42A mutant[34], particularly when expressed bivalently in an Fc-fusion format (Supplementary Fig. 1). To further reduce CD25 interactions, we next combined single mutations in a step-wise manner, first into double (Supplementary Figs 1–3), and then into triple mutants (Supplementary Fig. 4). Scanning mutagenesis using up to 14 different amino acid substitutions at targeted residues revealed strong positional effects and the requirement for at least three mutations in the interface to abolish activation of CD25$^+$ cells both *in vitro* and *in vivo* (Supplementary Figs 4 and 5).

We then benchmarked the activity of a novel IL-2$^{3X}$Fc triple mutant (R38D, K43E, E61R; Fig. 1a) against IL-2$^{WT}$Fc and IL-2/mAb immune complexes consisting of human IL-2 and the mouse anti-human antibody MAB602 (see Methods). Single-dose IL-2$^{3X}$Fc induced robust expansion of MP CD8 and NK cell subsets in the spleens of C57BL/6 mice, substantially higher than what was observed not only for IL-2$^{WT}$Fc, but also for treatment with IL-2/mAb immune complexes (Fig. 1b). The superior activity of mutant IL-2 fusion protein in a single-dose setting was consistent with a prolonged serum half-life relative to IL-2/mAb complexes and IL-2$^{WT}$Fc (Supplementary Figs 3E and 8). Notably, IL-2$^{3X}$Fc administration induced minimal expansion of Tregs confirming its high level of selectivity for CD8 and NK subsets (Fig. 1b).

IL-2$^{3X}$Fc also drove potent expansion of cytotoxic subsets in multiple low-dose treatments, similar to what was observed for IL-2/mAb, but substantially higher than the parental IL-2$^{WT}$Fc protein (Fig. 1c). However, while IL-2/mAb complexes caused considerable Treg expansion ($\sim 5$-fold), treatment with multiple low doses of IL-2$^{3X}$Fc failed to induce Treg expansion (Fig. 1c).

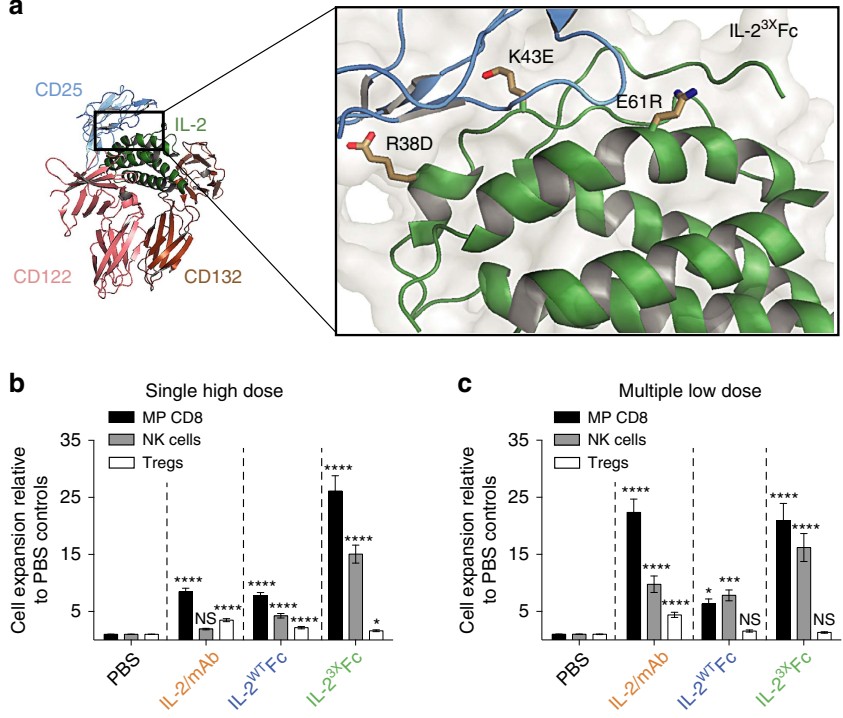

**Figure 1 | Abolition of CD25 binding results in potent and selective expansion of cytotoxic lymphocyte subsets. (a)** Selected charge-reversal mutations were introduced into human IL-2 based on inspection of the IL-2/IL-2R co-crystal structure[32] (PDB: 2B51), with the aim of disrupting CD25 binding. Highlighted are the mutations introduced to generate the IL-2$^{3X}$Fc triple mutant. R38D and E61R were selected after binding kinetics and cell-based assays (Supplementary Figs 1 and 2), while K43E was incorporated after screening of *in vivo* activity (Supplementary Fig. 4A–G). **(b,c)** Lymphocyte expansion profiles in the spleens of C57BL/6 mice receiving IL-2/mAb, IL-2$^{WT}$Fc or IL-2$^{3X}$Fc treatment. **(b)** Fold expansion in the total numbers of memory-phenotype CD8$^+$ T-cells (CD8$^+$ CD44$^{high}$ CD122$^{high}$), NK cells (CD3$^-$ NK1.1$^+$ CD122$^{high}$) and Tregs (CD4$^+$ FoxP3$^+$ CD25$^+$) after a single IL-2/mAb (3 µg IL-2 + 15 µg mAb) or IL-2-Fc (16.8 µg) i.p. injection was determined by flow cytometry on day 5. Shown is pooled data from seven independent experiments, each normalized to the average subset numbers of two to three PBS-treated mice (PBS, n = 16; treatment groups, n = 4–8). **(c)** Fold expansion in the total numbers of MP CD8, NK cells and Tregs after five IL-2/mAb (1 µg IL-2 + 5 µg mAb) or IL-2-Fc (5.6 µg) i.p. injections (days 0–4) was determined by flow cytometry on day 5. Shown is pooled data from five independent experiments, each normalized to the average subset numbers of two to three PBS-treated mice (PBS, n = 18; treatment groups, n = 8). Data are displayed as mean ± s.e.m. Asterisks indicate significant differences relative to PBS controls (*$P < 0.05$, ***$P < 0.001$, ****$P < 0.0001$) as determined by one-way analysis of variance with Bonferroni *post hoc* test for multiple comparisons.

Taken together, these experiments demonstrated that our design objectives had been achieved; with IL-2$^{3X}$Fc treatment causing prominent expansion of CD25$^-$ MP CD8 and NK cells but no expansion of CD25$^+$ Tregs.

**Low toxicity and potent antitumour activity of IL-2$^{WT}$Fc.** Having successfully designed a highly active and selective IL-2$^{3X}$Fc triple mutant we proceeded to evaluate its therapeutic potential. First, we examined mice for signs of treatment-associated toxicity. Notably, multiple injections of IL-2$^{3X}$Fc resulted in weight loss, suggesting that this variant induces systemic toxicity at the administered dose (Fig. 2a). Next, we assessed mice for pulmonary oedema and compromised hepatic function as a measure of experimentally induced vascular leak syndrome, a hallmark side effect of IL-2 therapy[13,35]. Treatment with IL-2/mAb or IL-2$^{3X}$Fc induced pulmonary oedema, as evidenced by increases in lung water content (Fig. 2b; Supplementary Fig. 6A). By contrast, lung water weight in mice treated with IL-2$^{WT}$Fc remained largely unchanged relative to PBS controls, either as absolute weight or as percentage of total body weight (Fig. 2b; Supplementary Fig. 6A). Assessment of liver weight and aspartate aminotransferase activity in serum revealed no differences to PBS controls across all treatment groups (Fig. 2c,e; Supplementary Fig. 6B). However, a significant elevation in alanine aminotransferase activity in serum

was observed in the IL-2/mAb group only (Fig. 2d; one-way analysis of variance, $P = 0.0065$). These low levels of hepatic toxicity may reflect the lower maximal serum concentrations of administered cytokine relative to experimental high-dose IL-2 (refs 14,35). Taken together, these results suggest a broad correlation between the magnitude of immune cell expansion and the development of treatment-associated toxicities, with low-dose IL-2/mAb or IL-2$^{3X}$Fc treatments displaying higher levels of toxicity, particularly pulmonary oedema, compared to IL-2$^{WT}$Fc.

We next evaluated the therapeutic efficacy of IL-2/mAb, IL-2$^{WT}$Fc and IL-2$^{3X}$Fc in the B16F10 melanoma model. For this purpose, we utilized a dosing regime consisting of five consecutive daily intraperitoneal (i.p.) injections of 1 µg antibody-complexed IL-2 or IL-2-Fc molar equivalent starting 1 day after subcutaneous injection of tumour cells. Strikingly, treatment with IL-2$^{WT}$Fc resulted in a substantial reduction of tumour growth compared to either IL-2/mAb immune complex or the IL-2$^{3X}$Fc triple mutant (Fig. 2f). This apparent superior therapeutic efficacy was observed despite the lower potential of IL-2$^{WT}$Fc to induce expansion of cytotoxic immune cell subsets (Fig. 1c).

**IL-2$^{WT}$Fc targets Tregs for FcγR-mediated depletion.** The observation that both IL-2/mAb immune complexes and the IL-2$^{3X}$Fc triple mutant displayed greater toxicity and less efficient

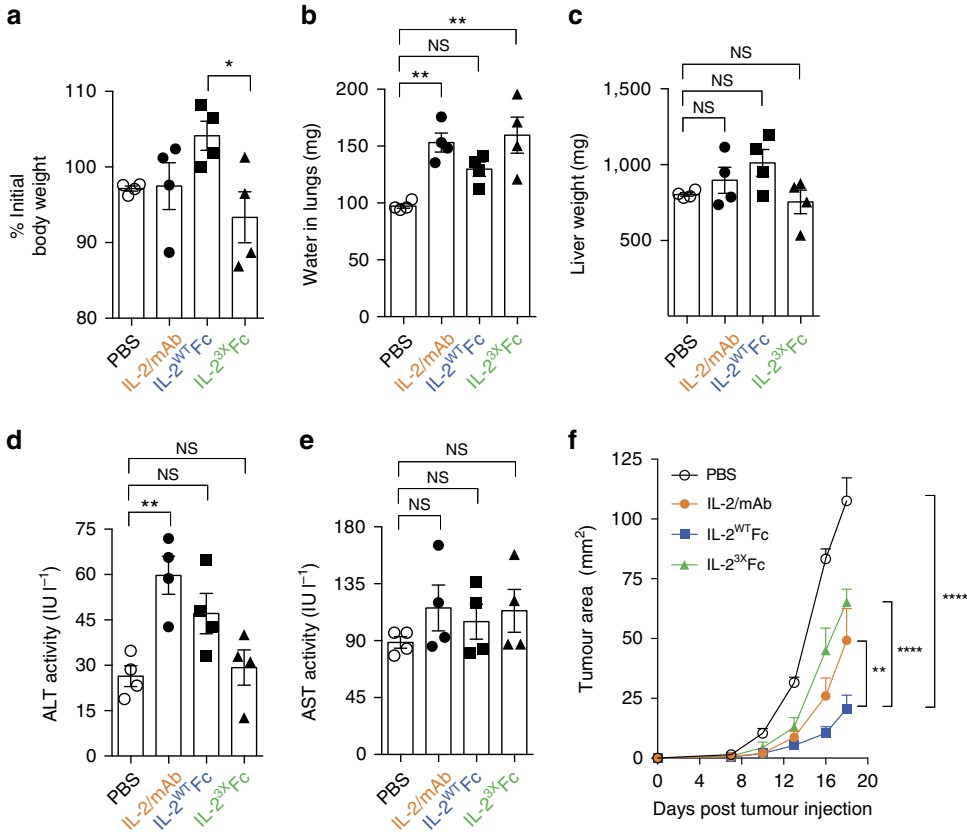

**Figure 2 | Toxicity profile and antitumour activity of IL-2-Fc variants.** (**a**–**e**) Symptoms of experimental VLS were assessed in mice receiving five consecutive doses (days 0–4) of IL-2/mAb (1 μg IL-2 + 5 μg mAb per dose), IL-2$^{WT}$Fc (5.6 μg per dose), IL-2$^{3X}$Fc (5.6 μg per dose) or PBS control. Lungs, livers and blood were collected on day 6 for analysis ($n = 4$ mice per group). (**a**) Body weight on day 6, represented as percentage of initial body weight on day 0. (**b**) Pulmonary oedema was assessed by measurement of lung water content, with significantly higher accumulation of fluid observed in the IL-2/mAb ($P = 0.0042$) and IL-2$^{3X}$Fc ($P = 0.0018$) groups, compared to PBS controls. (**c**–**e**) Assessment of liver toxicity as measured by total liver weight (**c**) and serum levels of liver enzymes alanine aminotransferase (ALT, **d**) and aspartate aminotransferase (AST, **e**). (**f**) Tumour growth after subcutaneous inoculation of B16F10 melanoma cells into the flanks of mice treated with five consecutive doses of IL-2/mAb (1 μg IL-2 + 5 μg mAb per dose), IL-2$^{WT}$Fc (5.6 μg per dose), IL-2$^{3X}$Fc (5.6 μg per dose) or PBS control on days 1–5 ($n = 6$). Data are displayed as mean ± s.e.m. Asterisks indicate significant differences between specified groups (*$P < 0.05$, **$P < 0.01$, ****$P < 0.0001$) as determined by one-way analysis of variance (ANOVA) (**a**–**e**) or two-way ANOVA (**f**) with Bonferroni *post hoc* test for multiple comparisons.

protection against tumour growth than wild-type IL-2-Fc-fusion protein was unexpected and led to further investigation. We focused our attention on the role of the Fc part of the molecule, and in particular its interaction with FcγRs. For this purpose, we investigated mutations that abolish antibody-dependent cell-mediated cytotoxicity/phagocytosis (ADCC/ADCP). More specifically, we mutated conserved residues in the Fc region of IL-2-Fc mediating binding to FcγRs (L234A, L235E and G237A)[36,37]. Accordingly, we assessed the activity of IL-2$^{WT}$Fc$^{nil}$ (no effector functions) and IL-2$^{WT}$Fc (normal FcγR binding but disrupted C1q interaction) constructs (Fig. 3a; Supplementary Fig. 7A). Since the Fc region utilized in this study was deficient for binding to the C1q complement component, we also generated a mutant (IL-2$^{WT}$Fc$^{C1q+}$) in which C1q binding was restored through mutation (see Methods). Disruption of the FcγR and/or C1q binding sites was validated through macrophage and C1q binding assays, respectively (Supplementary Fig. 7C,D). Importantly, mutation of the Fc region did not affect interaction with the CD25$^{high}$ CTLL-2 cell line (Supplementary Fig. 7B), indicating that the IL-2 component of IL-2-Fc remained intact. This allowed us to assign any differences in IL-2-Fc *in vivo* activity to the mutations introduced into the Fc component.

*In vivo* assessments in C57BL/6 mice receiving multiple low-dose IL-2-Fc treatment revealed similar increases in spleen lymphoid cellularity for all designed variants (Fig. 3b). Notably, the effector-less IL-2$^{WT}$Fc$^{nil}$ fusion protein readily expanded not only CD122$^{high}$ MP CD8 and NK cells but also CD4$^+$ CD25$^+$ Tregs (Fig. 3c,d). By contrast, treatment with IL-2$^{WT}$Fc and IL-2$^{WT}$Fc$^{C1q+}$ led to comparable increases in MP CD8 and NK cells, but no increase in the numbers or percentages of Tregs (Fig. 3c,d). In view of the similar levels of MP CD8 and NK cell expansion across all constructs and the lack of Treg expansion in the presence of Fc-mediated effector functions, we concluded that IL-2$^{WT}$Fc and IL-2$^{WT}$Fc$^{C1q+}$ were able to selectively deplete Tregs. We should emphasize, however, that this depletive effect was quite limited relative to PBS controls (Fig. 3e,f) and only became prominent when compared with the marked Treg expansion induced by IL-2$^{WT}$Fc$^{nil}$. Hence the Fc-mediated depletion seemed to be largely restricted to IL-2-activated Tregs. Furthermore, Treg depletion was predominantly FcγR-mediated rather than C1q-mediated since the ability to bind C1q (IL-2$^{WT}$Fc$^{C1q+}$) did not result in increased elimination of Tregs (Fig. 3c,d). Notably, a large proportion of Tregs in mice treated with IL-2$^{WT}$Fc or IL-2$^{WT}$Fc$^{C1q+}$ displayed high levels of the cell proliferation factor Ki-67 (Supplementary Fig. 7E). This finding

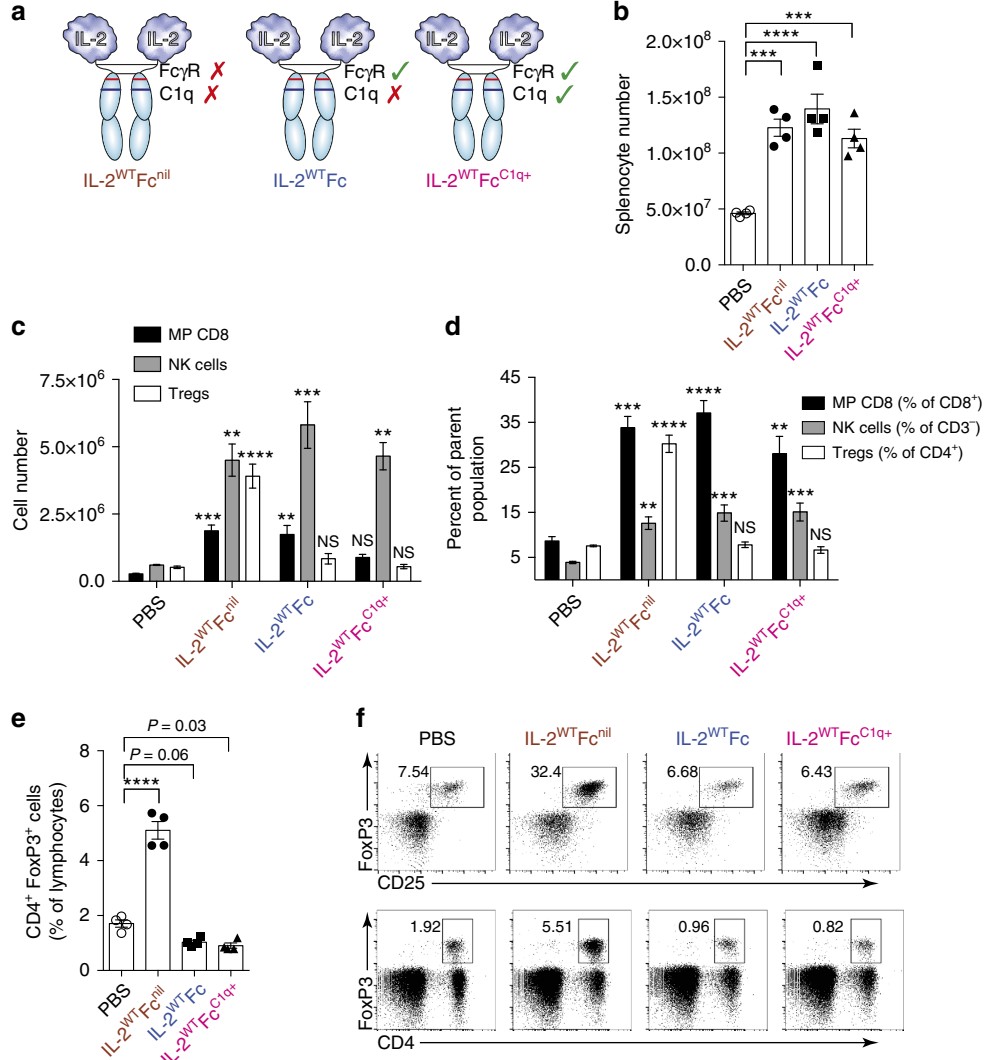

**Figure 3 | IL-2^WTFc selectively depletes Tregs in a FcγR-dependent manner.** (**a**) Diagram of produced IL-2-Fc variants showing FcγR and C1q-binding site status. Mutated residues within the Fc region are illustrated in Supplementary Fig. 7A. (**b–f**) Analysis of spleens (day 5) collected from C57BL/6 mice treated with five consecutive 5.6 µg i.p. injections of IL-2$^{WT}$Fc$^{nil}$, IL-2$^{WT}$Fc, IL-2$^{WT}$Fc$^{C1q+}$ or PBS control on days 0–4 (n = 4). (**b**) Total live splenocytes counts showing increased lymphoid cellularity after IL-2-Fc treatment. (**c–f**) Flow cytometric analysis of collected splenocytes. (**c**) Total cell numbers of MP CD8 (CD8$^+$ CD44$^{high}$ CD122$^{high}$), NK cell (CD3$^-$ NK1.1$^+$ CD122$^{high}$) and Treg (CD4$^+$ FoxP3$^+$ CD25$^+$) cell subsets. (**d**) Frequencies of MP CD8 (shown as proportion of CD8$^+$), NK cells (proportion of CD3$^-$) and Tregs (proportion of CD4$^+$). (**e**) Frequencies of CD4$^+$ FoxP3$^+$ cells within the lymphocyte compartment showing depletion of this subset after treatment with FcγR-binding IL-2-Fc constructs. (**f**) Representative flow cytometry dot plots displaying the frequency of regulatory T-cells in treated mice as defined by the co-expression of CD4, FoxP3 and the IL-2-inducible CD25 surface marker (top row, shown as proportion of CD4$^+$) or by expression of CD4 and FoxP3 (bottom row, shown as proportion of total lymphocytes). Data are displayed as mean ± s.e.m. Asterisks indicate significant differences relative to PBS controls (**c,d**) or between specified groups (**b,e**) as determined by one-way analysis of variance with Bonferroni *post hoc* test for multiple comparisons (**P < 0.01, ***P < 0.001, ****P < 0.0001).

supports the notion that these FcγR-binding constructs did stimulate Tregs to divide but also eliminated a large proportion of these cells via Fc-mediated killing, resulting in little or no change in Treg numbers.

We further investigated the observed preferential depletion of Tregs by assessing the interaction of IL-2-Fc with different immune cell subsets. Evaluation in an *ex vivo* binding assay revealed that IL-2$^{WT}$Fc preferentially bound CD4$^+$ CD25$^+$ Tregs over the CD25$^-$ MP CD8 or NK cell subsets (Fig. 4a). To explore differential targeting of lymphocyte subsets *in vivo*, we compared the cellular biodistribution profiles of fluorescently labelled IL-2$^{WT}$Fc and IL-2$^{WT}$Fc$^{nil}$ fusion proteins (Fig. 4b). We found that IL-2-Fc proteins targeted a low proportion of total CD8$^+$ T-cells, which is consistent with only a small fraction of

this compartment expressing high levels of the dimeric IL-2Rβγ$_c$ (that is, MP CD8 cells). Similarly, ∼12% of total CD4$^+$ T-cells bound to IL-2-Fc proteins, in agreement with typical proportions of Tregs expressing the trimeric IL-2Rαβγ$_c$. Accordingly, IL-2-Fc fusions were found to efficiently target CD25$^+$ FoxP3$^+$ Tregs (>85%, Fig. 4b), despite reductions in IL-2-Fc fluorescence intensity after sample fixation for FoxP3 immunostaining (Supplementary Fig. 5C). We observed that labelled IL-2-Fc proteins, regardless of their ability to bind FcγR, targeted ∼95% of NK cells, thus suggesting that these constructs bind to this subset predominantly through the IL-2R (IL-2Rβγ$_c$) rather than via FcγR. By contrast, disruption of FcγR binding substantially reduced the proportion of macrophages and neutrophils associated with IL-2$^{WT}$Fc$^{nil}$ (Fig. 4c), suggesting that

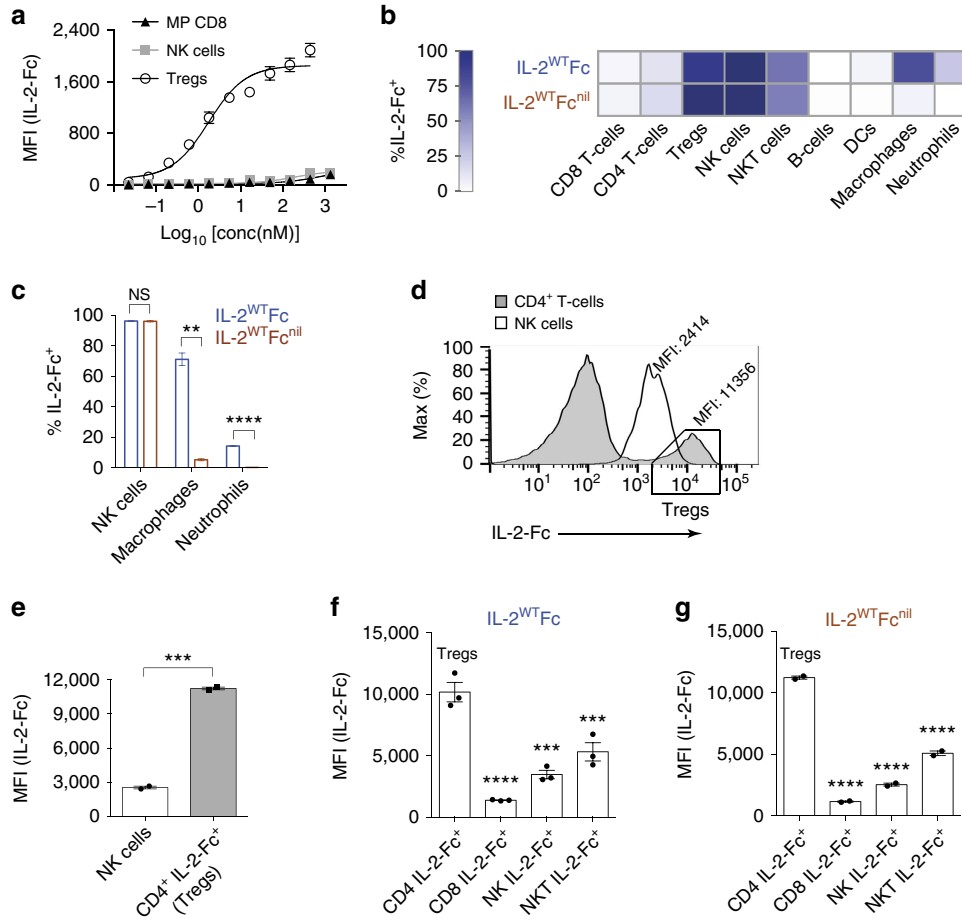

**Figure 4 | Depletive IL-2^{WT}Fc activity relies on high-affinity targeting of Tregs and interaction with myeloid effector subsets.** (**a**) *Ex vivo* flow cytometric detection of labelled IL-2^{WT}Fc on the surface of MP CD8, NK cell and Treg subsets after incubation with Fc-blocked *FoxP3*^{DTR/GFP} splenocytes ($n = 2$ technical replicates). (**b–g**) Cellular biodistribution profiles of fluorescently labelled IL-2^{WT}Fc and IL-2^{WT}Fc^{nil} in the spleens of treated C56BL/6 mice as determined by flow cytometry 12 h post injection (16.8 μg IL-2-Fc i.p., $n = 2–3$ mice per group). (**b**) Heat-map representation of the percentages of lymphoid and myeloid subsets bound by IL-2-Fc-fusion proteins (**c**) Abolition of FcγR binding causes a significant reduction in the percentages of macrophages ($P = 0.0041$) and neutrophils ($P < 0.0001$) bound by fluorescent IL-2-Fc, as determined by two-tailed unpaired Student's *t*-test. (**d**) Representative histograms displaying the levels of IL-2^{WT}Fc^{nil} present on CD4^+ T-cells and NK cells after i.p. injection. Box indicates that the majority of CD4^+ IL-2-Fc^+ cells are Tregs, as previously shown in Supplementary Fig. 5C. (**e**) Quantification of **d**, showing significantly higher IL-2-Fc MFI values in IL-2-Fc^+ CD4^+ T-cells (boxed cells in **d**) relative to IL-2-Fc^+ NK cells ($n = 2$ mice, two-tailed unpaired Student's *t*-test, $P = 0.0004$). (**f,g**) Injected IL-2^{WT}Fc (**f**) and IL-2^{WT}Fc^{nil} (**g**) proteins accumulate to higher levels on the surface of Tregs compared to any other analysed subset. Asterisks indicate significant differences relative to CD4^+ IL-2-Fc^+ Tregs, as determined by one-way analysis of variance with Bonferroni *post hoc* test for multiple comparisons (**$P < 0.01$, ***$P < 0.001$, ****$P < 0.0001$). All data are displayed as mean ± s.e.m. MFI, mean fluorescence intensity.

these myeloid subsets may function as effector cells in FcγR-dependent Treg depletion. Finally, similar frequencies of NKT cells (∼50%), dendritic cells (∼3%) and B-cells (∼1%) were found to be targeted by both IL-2^{WT}Fc and IL-2^{WT}Fc^{nil} proteins (Fig. 4b).

Having examined the frequencies of IL-2-Fc^+ cells, we next compared the intensity of IL-2-Fc fluorescent signals, specifically on the NK cell and Treg subsets. To exclude any potential for FcγR-mediated binding on NK cells, fluorescently labelled IL-2^{WT}Fc^{nil} was used for this comparison. Interestingly, the mean fluorescence intensity levels observed on IL-2-Fc^+ CD4^+ T-cells, of which the vast majority are Tregs (see Supplementary Fig. 5C), were nearly fivefold higher compared to IL-2-Fc^+ NK cells (Fig. 4d,e). Likewise, fluorescently labelled IL-2^{WT}Fc and IL-2^{WT}Fc^{nil} proteins were both observed at higher amounts on Tregs than on CD8^+ T-cells and NKT cells in addition to NK cells (Fig. 4f,g).

Collectively, our *ex vivo* (Fig. 4a) and *in vivo* (Fig. 4c–g) analyses demonstrate substantially stronger binding of IL-2-Fc proteins to high-affinity IL-2Rαβγ_c on Tregs than to CD25^- subsets expressing intermediate-affinity IL-2Rβγ_c, thus explaining the selective opsonization of Tregs by IL-2^{WT}Fc constructs.

**IL-2-Fc antitumour effects require CD25 and FcγR interaction.** To further investigate the influence of Fc-mediated effector functions on antitumour effects, we compared the efficacy of IL-2^{WT} and IL-2^{3X} proteins expressed as fusions to Fc (able to bind FcγR) or Fc^{nil} (abolished effector functions). As previously observed (Fig. 2f), treatment with IL-2^{WT}Fc resulted in superior antitumour activity against B16F10 melanoma in comparison to IL-2^{3X}Fc (Fig. 5a), despite the latter variant mediating considerably higher peripheral expansion of cytotoxic subsets (Fig. 1b,c). Remarkably, the efficacy of IL-2^{WT}Fc was critically

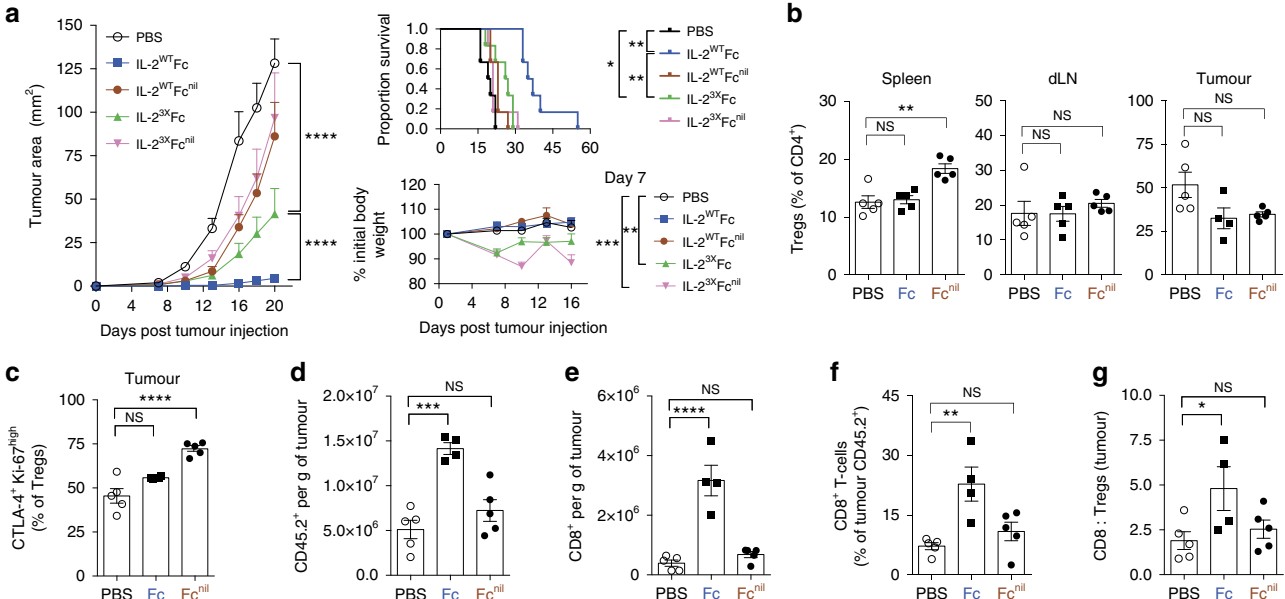

**Figure 5 | Potent antitumour activity of IL-2$^{WT}$Fc is dependent on both Fc$\gamma$R and CD25 binding.** (**a**) The antitumour activity of IL-2$^{WT}$ and IL-2$^{3X}$ fused to either Fc or Fc$^{nil}$ was assessed in the B16F10 melanoma tumour model ($n = 6$). C57BL/6 mice were injected subcutaneously (s.c.) in their left flanks with $1 \times 10^5$ B16F10 cells (day 0) and treated with five consecutive doses of IL-2-Fc variants (5.6 $\mu$g per dose, i.p.) on days 1–5. Following treatment, mice were monitored for tumour growth (left), survival (top right) and body weight (bottom right). (**b-g**) Flow cytometric analysis of spleens, dLNs (inguinal) and B16F10 tumours collected from mice receiving IL-2$^{WT}$Fc, IL-2$^{WT}$Fc$^{nil}$ or PBS control treatment ($n = 5$). Following s.c. tumour inoculation (day 0), mice received a total of ten doses of IL-2-Fc on days 1–5 and days 14–18 (5.6 $\mu$g per dose, i.p.), followed by analysis 48 h after the last dose (day 20). (**b**) Frequency of FoxP3$^+$ cells (percentage of CD4$^+$) in the spleen, dLNs and tumours. (**c**) Frequency of CTLA-4$^+$ Ki-67$^{high}$ intratumoural Tregs (percentage of CD4$^+$ FoxP3$^+$ cells). (**d-e**) Numbers of infiltrating CD45.2$^+$ leukocytes (**d**) and CD8$^+$ T-cells (**e**) per gram of tumour. (**f**) Frequency of tumour-infiltrating CD8$^+$ T-cells, shown as percentage of CD45.2$^+$ cells. (**g**) Intratumoural ratio of CD8$^+$ to regulatory T-cells, as calculated from their relative proportions in the CD45.2$^+$ compartment. dLNs, draining lymph nodes; one mouse in the IL-2$^{WT}$Fc group did not develop a tumour. Data are displayed as mean ± s.e.m. Asterisks indicate significant differences between specified groups (*$P < 0.05$, **$P < 0.01$, ***$P < 0.001$, ****$P < 0.0001$) as determined by one-way analysis of variance (ANOVA) (**b-g**) or two-way ANOVA (**a**) with Bonferroni *post hoc* test for multiple comparisons. Survival (**a**, top right) is displayed using Kaplan–Meier plots and compared by the Gehan–Breslow–Wilcoxon test.

dependent on Fc$\gamma$R binding, since treatment with IL-2$^{WT}$Fc$^{nil}$ resulted in no detectable differences in tumour growth or survival compared to PBS controls (Fig. 5a). Furthermore, as observed in non-tumour bearing mice (Fig. 2a), treatment with the IL-2$^{3X}$Fc variant caused notable reductions in body weight (Fig. 5a, bottom right). This effect was accentuated in mice receiving IL-2$^{3X}$Fc$^{nil}$ treatment, possibly due to the elevated levels of fusion protein detected in serum compared to IL-2$^{3X}$Fc (Supplementary Fig. 8).

Flow cytometric analyses of spleen, draining lymph node (dLN) and B16F10 tumour tissue were performed in order to gain insights into mechanisms underpinning the efficacy of IL-2$^{WT}$Fc treatment. For this purpose, we compared the levels of lymphocyte expansion in mice treated with either IL-2$^{WT}$Fc or IL-2$^{WT}$Fc$^{nil}$ relative to PBS control treatment 48 h after the last IL-2-Fc dose (Fig. 5b–g). In agreement with our previous results (Fig. 3), we found that IL-2$^{WT}$Fc$^{nil}$ treatment readily expanded splenic Tregs, while IL-2$^{WT}$Fc failed to expand this subset (Fig. 5b, left). While these results reinforce the previously observed Treg-depletive activity of IL-2$^{WT}$Fc in the spleen (Fig. 3), this effect was less pronounced in dLN (Fig. 5b, middle) and not evident in tumour lesions (Fig. 5b, right) at this time point. Notably, however, flow cytometric analysis performed 24 h after the last administered dose of fusion protein revealed that IL-2$^{WT}$Fc is indeed able to mediate Treg depletion in the spleen, dLN and, crucially, at the tumour site itself (Supplementary Fig. 9A,B).

We observed an increased proportion of CTLA-4$^+$ Ki-67$^{high}$ Tregs in the tumours of mice treated with IL-2$^{WT}$Fc$^{nil}$ but not with IL-2$^{WT}$Fc (Fig. 5c). Interestingly, this difference was tumour-specific and was not observed in the spleen or dLN (Supplementary Fig. 9C). Thus, the reduced frequency of highly activated CTLA-4$^+$ Ki-67$^{high}$ Tregs in the tumours of IL-2$^{WT}$Fc-treated mice may provide a basis for improved antitumour responses. In line with this observation, B16F10 tumours collected from mice receiving IL-2$^{WT}$Fc treatment displayed a pronounced increase in the numbers of infiltrating leukocytes (Fig. 5d) with a clear enrichment of cytotoxic CD8$^+$ T-cells, both in terms of numbers (Fig. 5e) and frequency (Fig. 5f). This enrichment translated into an increased intratumoural ratio of CD8$^+$ to Tregs in the IL-2$^{WT}$Fc treatment group only (Fig. 5g). In support of this observation, antibody-mediated depletion of CD8$^+$ T-cells severely compromised IL-2$^{WT}$Fc antitumour activity (Supplementary Fig. 9D), while depletion of total CD4$^+$ T-cells improved treatment efficacy (Supplementary Fig. 9E), presumably by further depletion of CD4$^+$ CD25$^+$ Tregs.

We next investigated the potential of the IL-2-Fc-fusion proteins for therapy in the B16F10 model in combination with tumour-targeting antibodies[30]. For this purpose IL-2/mAb, IL-2$^{WT}$Fc or IL-2$^{3X}$Fc treatment was administered in combination with a monoclonal antibody targeting the B16F10 tumour antigen tyrosinase-related protein 1 (TRP-1) (Fig. 6a). In this setting, IL-2$^{WT}$Fc again provided the largest reduction in tumour growth and was the only treatment to significantly improve survival compared to anti-TRP-1 monotherapy (Gehan–Breslow–Wilcoxon test, $P = 0.0446$).

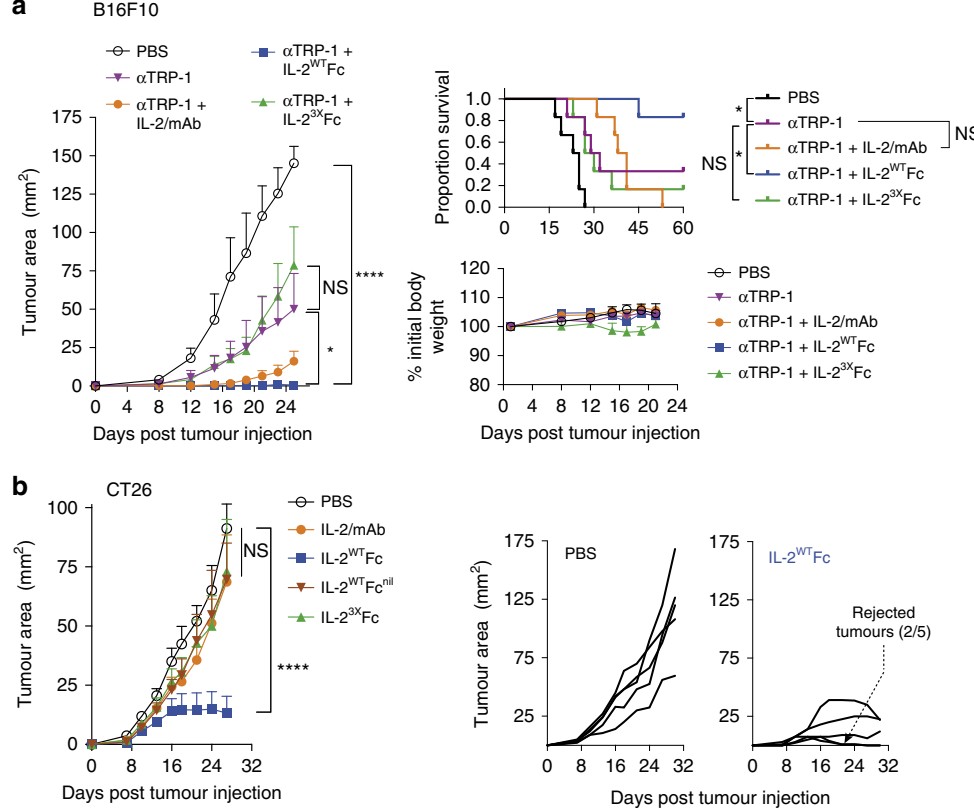

**Figure 6 | IL-2^WTFc treatment synergizes with targeted antibody therapy and is highly efficacious against murine colorectal carcinoma.** (a) The antitumour activity of IL-2/mAb, IL-2^WTFc and IL-2^3XFc in combination with the anti-TRP-1 tumour-targeting monoclonal antibody was assessed in the B16F10 melanoma tumour model. C57BL/6 mice were injected subcutaneously (s.c.) in their left flanks with $1 \times 10^5$ B16F10 cells on day 0 ($n = 6$). Combination therapy consisted of three doses of anti-TRP-1 (200 μg per dose given i.p. on days 3, 6 and 9) plus low-dose IL-2/mAb (0.5 μg IL-2 + 2.5 μg mAb per dose) or molar IL-2-Fc equivalent (2.8 μg per dose) given i.p. on days 4, 7 and 10. Following treatment, mice were monitored for tumour growth (left), body weight (bottom right) and survival (top right). (b) The antitumour activity of IL-2/mAb, IL-2^WTFc, IL-2^3XFc and IL-2^WTFc^nil was assessed in the CT26 murine colorectal carcinoma tumour model. Balb/c mice were injected s.c. in their left flanks with $1 \times 10^5$ CT26 cells on day 0 ($n = 5$), followed by treatment with a total of nine doses of IL-2/mAb (1 μg IL-2 + 5 μg mAb per dose), IL-2-Fc molar equivalent (5.6 μg per dose) or PBS control given i.p. on days 1–5 and days 13, 15, 17 and 19. Mean tumour size (left) and tumour growth in individual mice in the PBS and IL-2^WTFc groups (right) are shown. Data are displayed as mean ± s.e.m. Asterisks indicate significant differences between specified groups (*$P < 0.05$, ****$P < 0.0001$) as determined by two-way analysis of variance with Bonferroni *post hoc* test for multiple comparisons. Survival (a, top right) is displayed using Kaplan–Meier plots and compared by the Gehan–Breslow–Wilcoxon test.

In addition to B16F10 melanoma, the efficacy of IL-2-Fc-fusion proteins was also investigated in the CT26 colorectal carcinoma syngeneic tumour model (Fig. 6b). Results in this model closely resembled what had been observed for B16F10, with IL-2^WTFc providing substantial reductions in tumour growth, while all other treatments displayed no detectable effects. Remarkably, some of the mice treated with IL-2^WTFc displayed complete rejection of subcutaneous tumours (Fig. 6b, right), further highlighting the potential of IL-2^WTFc therapy.

## Discussion

Considered the first effective cancer immunotherapy, high-dose IL-2 is able to mediate durable responses in a small subset of metastatic melanoma and metastatic renal cancer patients[6]. However, IL-2 suffers from sub-optimal therapeutic properties arising from a short serum half-life and pleiotropic biological actions, which lead to toxicity and the unwanted expansion of immunosuppressive Tregs.

Here we have generated highly selective IL-2-Fc-fusion proteins through introduction of CD25-disrupting mutations into the cytokine component. Using a combination of structural design,

*in vitro* biosensor measurements and *in vivo* screening, we developed a novel IL-2-Fc triple mutant (IL-2^3XFc). Disruption of CD25 binding resulted in prolongation of serum half-life, large increases in biological activity and preferential expansion of CD25^− CD122^high MP CD8 and NK cells. Notably, IL-2^3XFc displayed enhanced activity and selectivity compared to not only wild-type IL-2-Fc, but also previously reported IL-2/mAb complexes.

Further analyses revealed that the antitumour effects and toxicity profile of the IL-2^3XFc triple mutant were comparable to that of IL-2/mAb complexes, indicating that our initial design objectives had been achieved. However, intriguingly, these experiments also revealed that both IL-2^3XFc and IL-2/mAb were in fact less effective than the parental IL-2^WTFc protein in the B16F10 melanoma model. Furthermore, unlike IL-2^3XFc and IL-2/mAb treatment, IL-2^WTFc did not induce pulmonary oedema or reductions in body weight. This indicated that toxicities arising from these low-dose treatments might be largely immune-related, likely due to excessive activation of CD8^+ T-cells and NK cells (although residual effects on the lung endothelium could not be excluded)[13].

Subsequent analyses revealed that selective Treg depletion was critically dependent on the high affinity of IL-2^WTFc towards

$CD25^+$ cells, as well as the interaction of the Fc region with FcγR. Crucially, the potent antitumour activity displayed by $IL-2^{WT}Fc$ was also dependent on these two properties, with mutation of either the CD25- or FcγR-binding sites resulting in reduced treatment efficacy in the B16F10 melanoma and CT26 colorectal carcinoma models. Analysis of B16F10 tumours revealed reduced frequencies of CTLA-4$^+$ Ki-67$^{high}$ Tregs after $IL-2^{WT}Fc$ treatment, as well as a large increase in total leukocyte infiltration dominated by cytotoxic $CD8^+$ T-cells. Our results therefore suggest that $IL-2^{WT}Fc$ administration contributes to the establishment of a highly immunogenic tumour microenvironment that favours the infiltration of immune cells in general, and of $CD8^+$ T-cells in particular, leading to enhanced antitumour responses. This is consistent with previously reported increases in $CD8^+$ T-cell tumour infiltration and subsequent tumour rejection following *in vivo* Treg depletion in transgenic mouse models[38]. The data are also in agreement with the finding that, in the context of immune checkpoint blockade, effective antitumour immunity correlates with a high ratio of $CD8^+$ T-cells to Tregs in tumour-infiltrating lymphocytes[39,40].

In addition to the depletion of Tregs, $IL-2^{WT}Fc$ treatment also resulted in expansion of splenic MP CD8 and NK cells, although to a lesser extent than what was observed for $IL-2^{3X}Fc$ or IL-2/mAb complexes. The provision of dual activities, which can be either depleting or proliferative depending on the cellular subset, differentiates $IL-2^{WT}Fc$ from existing therapeutic modalities, such as the anti-CD25 monoclonal antibody daclizumab (Zenapax)[41]. This property may also provide advantages over IL-2-toxin fusions, which target Tregs for depletion but also mediate collateral elimination of $CD8^+$ T-cells and NK cells[42,43]. Furthermore, pre-administration of $IL-2^{WT}Fc$ could be used to improve the tumour-targeting properties of IL-2 immunocytokines, as recently reported by Wittrup and colleagues[29]. Finally, based on our findings, ultra-low doses of the highly active and selective $IL-2^{3X}Fc$ variant may prove effective in combination with Treg-depleting antibodies, a strategy that we are currently evaluating.

Taken together, our results on the engineering of IL-2-Fc-fusion proteins outline an effective strategy to enhance the therapeutic properties of this key immunomodulatory cytokine. Recent clinical successes of immune checkpoint inhibitors highlight the potential of novel immune modulating agents in cancer immunotherapy, either as stand-alone therapeutics or in combination with anti-PD-1 and anti-CTLA-4 therapy[44]. The $IL-2^{WT}Fc$ fusion protein described here displays favourable properties, as characterized by a long serum half-life, low levels of toxicity, increased antitumour activity and the ability to selectively deplete Tregs. The role of Treg depletion in the IL-2 system had so far remained elusive: indeed, IL-2/mAb complexes have been reported to act independently of Fc-mediated effector functions[14] and previous reports of IL-2-Fc-fusion proteins have failed to identify any evidence of Treg depletion[25,27–29]. By contrast, such a mechanism has been apparent for other targets and therapeutic modalities. In particular, FcγR-mediated Treg depletion has been recently described as a critical component in the antitumour activity of antibodies targeting CTLA-4, OX40 and GITR[45–47].

In summary, here we describe the development and characterization of IL-2-Fc-fusion proteins that exceed the specificity ($IL-2^{3X}Fc$) and efficacy ($IL-2^{WT}Fc$) of IL-2/mAb complexes. Moreover, we identify the depletion of Tregs, rather than the expansion of cytotoxic $CD8^+$ T-cells and NK cells, as a major determinant for antitumour activity, providing important guidance for the future development of IL-2 reagents for cancer immunotherapy.

## Methods

**Mutagenesis and production of recombinant proteins.** For the generation of IL-2-Fc-fusion proteins, regions encoding human IL-2 (residues Ala1–Thr133)

were genetically fused to murine Fc (IgG2c Glu216–Lys447, Eu antibody numbering[48]) containing a mutated C1q-binding site (E318A, K320A and K322A)[49] by means of a short glycine-serine linker (GSGS). The construct was generated by gene synthesis (GeneArt) and cloned into the mammalian expression vector pCEP4 (Thermo Fisher). Disruption of the CD25 and FcγR binding sites and restoration of the C1q binding site were performed using the Q5 site-directed mutagenesis kit (NEB) using custom-designed primers, followed by validation of mutations by Sanger sequencing. IL-2-Fc constructs were used to transfect suspension-adapted HEK293 cells using the Expi293 expression system (Thermo Fisher). Protein purification was performed with protein G agarose beads (ACROBiosystems) using disposable columns (Thermo Fisher). All protein preparations were quality controlled for endotoxin levels, as measured by chromogenic LAL assay (Lonza). Genetic constructs coding for the ectodomains of hCD25 and hCD122 (in pCEP4, C-terminal His-tagged) were purchased from Genscript, expressed as above and purified using the TALON metal affinity resin (Clontech).

**Mice.** C57BL/6 and BALB/c female mice were purchased from the Animal Resources Centre (Canning Vale, WA, Australia) and used at 8–10 weeks of age. IL-7 transgenic mice (IL-7 Tg: B6 background, Thy-1.1-congenic) and transgenic mice expressing DTR-GFP under the FoxP3 promoter[50] (C57BL/6 background) were bred at the Australian BioResources facilities (Moss Vale, NSW, Australia). Animals were housed under conventional barrier protection and handled in accordance with protocols approved by the Garvan Institute of Medical Research and St Vincent's Hospital Animal Experimentation and Ethics Committee, which comply with the Australian code of practice for the care and use of animals for scientific purposes.

**Flow cytometry and antibodies.** Flow cytometric analysis of mouse spleens and lymph nodes was performed according to standard protocols. The following antibodies were used for staining (all purchased from eBioscience unless stated otherwise): PE- or eFluor450-conjugated anti-CD8a (clone 53-6.7, used at 1.25 μg ml$^{-1}$); FITC- or eFluor450-conjugated anti-CD122 (clone TM-b1, 1 μg ml$^{-1}$); APC-conjugated anti-CD44 (clone IM7, 0.4 μg ml$^{-1}$); FITC- or eFuor450-conjugated anti-CD45.2 (clone 104, 2.5 μg ml$^{-1}$); PE-Cy7- or eFluor450-conjugated anti-CD3e (clone145-2C11, 1 μg ml$^{-1}$); PE- or APC-conjugated anti-NK1.1 (clone PK136, 1 μg ml$^{-1}$); BV421- or BV605-conjugated anti-CD4 (clone GK1.5, 1 μg ml$^{-1}$, BioLegend); PerCP/Cy5.5-conjugated anti-Thy-1.1 (clone HIS51, 0.5 μg ml$^{-1}$); PerCP/Cy5.5- or eFluor450-conjugated anti-CD25 (clone PC61.5, 1 μg ml$^{-1}$); PerCP-conjugated anti-B220 (clone RA3-6B2, 1 μg ml$^{-1}$, BioLegend); PE/Cy7-conjugated anti-Ly6G (clone 1A8, 1 μg ml$^{-1}$, BD); PE-conjugated anti-CD11b (clone M1/70, 1 μg ml$^{-1}$); PerCP-conjugated anti-CD11c (clone N418, 1 μg ml$^{-1}$, BioLegend); FITC-labelled anti-F4/80 (clone BM8, 5 μg ml$^{-1}$, BioLegend); PE-, APC- or eFluor450-conjugated anti-FoxP3 (clone FJK-16 s, 5 μg ml$^{-1}$); FITC- or PE-conjugated anti-Ki-67 (clone SolA15, 0.4 μg ml$^{-1}$) and APC-conjugated anti-CTLA-4 (clone UC10-4F10-11, 1 μg ml$^{-1}$, BD). Intracellular staining with anti-FoxP3, anti-Ki-67 and anti-CTLA-4 antibodies was performed after fixation/permeabilization with the FoxP3 buffer set (eBioscience).

**Cytokine treatments.** Mice received recombinant hIL-2, IL-2/mAb or IL-2-Fc treatments at specific quantities and dosing schedules, as described in the figure legends. Recombinant hIL-2 was obtained from Peprotech. IL-2/mAb complexes were prepared by mixing hIL-2 and the anti-hIL-2 MAB602 (mouse IgG2a, clone 5355, R&D systems) at a 2:1 molar ratio (for example, 1 μg IL-2 + 5 μg mAb), followed by incubation at 37 °C for 25 min prior to injection. Purified IL-2-Fc variants were stored at −80 °C, thawed, filtered (0.22 μm) and re-assessed for protein concentration prior to injection.

**Assessment of treatment-associated toxicity.** Mice were injected i.p. once per day with PBS, 1 μg IL-2 + 5 μg mAb, 5.6 μg $IL-2^{WT}Fc$ or 5.6 μg $IL-2^{3X}Fc$ on days 0–4. On day 6, mice were killed for collection of blood (cardiac puncture), lungs, livers and spleens. Blood samples were allowed to clot for 2 h at room temperature (RT), followed by separation of serum by centrifugation. Aspartate aminotransferase and alanine aminotransferase activity assays (Teco Diagnostics) were performed on non-hemolyzed serum samples as per manufacturer's instructions. Tissue wet weights were recorded using an analytical balance. To determine lung water content, lungs were dehydrated overnight at 42 °C using a SpeedVac instrument (Savant) and the difference between lung wet weight and lung dry weight was calculated.

**Assessment of IL-2-Fc binding to mouse splenocytes ex vivo.** Red blood cell-depleted splenocytes from a FoxP3-DTR/GFP transgenic mouse were incubated with Fc-Block (anti-mouse CD16/CD32, BD), followed by washing in flow cytometry buffer and staining with fluorophore-conjugated anti-CD3e, anti-CD8a, anti-CD4, anti-CD44 and anti-NK1.1 antibodies. Stained cells were washed, seeded in a 96-well plate ($1 \times 10^6$ cells per well) and re-suspended in serially diluted $IL-2^{WT}Fc$-biotin. After incubation for 30 min on ice, cells were

stained with APC-conjugated streptavidin (eBioscience). The levels of IL-2$^{WT}$Fc bound to the surface of MP CD8 (CD3$^+$ CD8$^+$ CD44$^{high}$), NK cells (CD3$^-$ NK1.1$^+$) and Tregs (CD3$^+$ CD4$^+$ FoxP3$^+$) were determined by flow cytometry.

**IL-2-Fc fluorescent labelling and cellular biodistribution.** Purified IL-2-Fc variants in PBS were filtered through a syringe-driven 0.22 μm filter prior to labelling with Alexa Fluor 647-NHS (Thermo Fisher). IL-2-Fc concentration was adjusted to 300 μg ml$^{-1}$ in 500 μl, followed by addition of 50 μl 1 M sodium bicarbonate to increase pH. Alexa Fluor 647-NHS was added at a 15:1 molar ratio and labelling was allowed to take place for 2 h at RT with gentle rotation in the dark. After the labelling reaction, IL-2-Fc variants were buffer-exchanged into PBS using two sequential de-salting steps with Zeba spin columns (Thermo Fisher). Final protein concentration and labelling efficiency were measured using a Nanodrop instrument (Thermo Fisher). Fluorescently labelled IL-2-Fc variants were administered to mice via i.p. as a single 16.8 μg dose and spleens were collected 12 h post injection. Flow cytometric analysis of red blood cell-depleted splenocytes was performed after staining with antibodies conjugated to fluorophores with different emission spectra to that of Alexa Fluor 647. The frequencies and mean fluorescence intensity levels of IL-2-Fc$^+$ cells were determined for the following populations: CD8$^+$ T-cells (CD3$^+$ CD8$^+$), CD4$^+$ T-cells (CD3$^+$ CD4$^+$), Tregs (CD4$^+$ FoxP3$^+$ CD25$^+$), NK cells (CD3$^-$ NK1.1$^+$), NKT cells (CD3$^+$ NK1.1$^+$), B-cells (CD3$^-$ B220$^+$), dendritic cells (CD3$^-$ CD11c$^{high}$), macrophages (CD3$^-$ CD11b$^{mid-high}$ CD11c$^{low-mid}$ F4/80$^+$ SSC$^{low}$) and neutrophils (CD3$^-$ CD11b$^{mid-high}$ CD11c$^{low-mid}$ Ly6G$^+$).

**Tumour models.** B16F10 melanoma cells (ATCC CRL-6475 Lot 60508145) were cultured at 37 °C, 5% CO$_2$ in high-glucose DMEM containing 10% FBS, 2 mM L-glutamine, 50 U ml$^{-1}$ penicillin and 50 μg ml$^{-1}$ streptomycin. B16F10 cells (70–80% confluent) were reconstituted in PBS and injected subcutaneously into the left flanks of C57BL/6 mice (1 × 10$^5$ cells per mouse). CT26 colorectal carcinoma cells (ATCC CRL-2638 Lot 63226308) were cultured in ATCC-formulated RPMI 1640 supplemented with 10% FBS, 50 U ml$^{-1}$ penicillin and 50 μg ml$^{-1}$ streptomycin. CT26 cells (70–80% confluent) were reconstituted in PBS and injected subcutaneously into the left flanks of BALB/c mice (1 × 10$^5$ cells per mouse). After tumour injection, mice were treated with specific IL-2/mAb, IL-2-Fc and anti-TRP-1 (clone TA99, BioXCell) dosing schedules, as described in the figure legends. Following treatment, tumour area was monitored every 2–3 days using callipers. Mice were killed if showing considerable weight loss (>20% of initial body weight), displayed obvious signs of systemic illness or if tumours grew larger than 144 mm$^2$. Mycoplasma-tested B16F10 and CT26 cancer cell lines were purchased from the American Type Culture Collection (ATCC), expanded in culture once and aliquoted for storage in liquid nitrogen. Aliquots were then thawed, expanded for use in a single tumour model experiment and discarded.

**Flow cytometric analysis of B16F10 tumours.** B16F10 tumours were dissected and carefully separated from skin tissue. Collected tumours were weighted and collected into serum-free RPMI 1640 (Thermo Fisher) containing 2 mg ml$^{-1}$ collagenase D (Roche) plus 0.1 mg ml$^{-1}$ DNAse I (grade I, Roche) and digested for 1 h at 37 °C. Tissue was then dispersed through 70 μm cell strainers (BD) to obtain single cell suspensions. The volume of cell suspension utilized for immunostaining was recorded for each sample (volume equivalent to ~20 mg), followed by aliquoting into a 96-well plate and addition of 5 × 10$^4$ counting beads (FITC Calibrite, BD) per well. Samples were then incubated with Fc-Block (anti-mouse CD16/CD32, BD), surface antibody stain and fixable viability dye (eFluor780, eBioscience) in that order. This was followed by fixation/permeabilization (FoxP3 buffer set, eBioscience) and incubation with intracellular stain. Samples were then analysed by flow cytometry, as described above. Total numbers of immune subsets per gram of tumour were calculated utilizing the following formula: [(no. of acquired cells÷no. of acquired beads) × no. of added beads]÷[(volume stained÷total volume) × tumour weight in grams].

**Affinity measurements.** Biolayer interferometry (BLI) measurements were performed using the BLItz instrument (ForteBio). hCD25-Fc (R&D systems) was biotinylated with EZ-Link NHS-PEG4-Biotin (Thermo Fisher) and loaded onto streptavidin biosensors. Data were obtained using IL-2-Fc variants at 100 μg ml$^{-1}$ (1,190 nM), with 120 s association and 600 s dissociation times. Surface plasmon resonance measurements were performed using the Biacore 2000 system. Recombinant hCD25-His, hCD122-His or mCD25 (R&D systems) were covalently immobilized onto CM5 sensor chips (GE) to ~200 response units using the amine coupling kit (GE). Binding of IL-2-Fc variants to hCD25, hCD122 or mCD25 was recorded using a 30 μl min$^{-1}$ flow rate, with 60 s of association and 300 s of dissociation followed by regeneration of immobilized ligands in 0.1 M glycine, 0.1 M NaCl pH 3. Curve fitting of hCD25 and mCD25 kinetic data was performed using the BIAevaluation software. One-armed IL-2$^{WT}$Fc was produced in order to validate the curve fitting strategy used for bivalent IL-2-Fc variants. Briefly, IL-2$^{WT}$Fc and Fc-His constructs (both in pCEP4 expression vector) were co-transfected into Expi293 cells (Thermo Fisher), followed by purification of the

monovalent IL-2$^{WT}$Fc/Fc-His pairing with a HisTrap column (GE) using an imidazole gradient elution (AKTA protein purification system, GE).

**Assessment of binding and activity on CTLL-2 cells.** The murine CTLL-2 cell line (IL-2-dependent, CD25$^{high}$) was cultured in complete RPMI 1640 (10% FBS, 2 mM L-glutamine, 50 U ml$^{-1}$ penicillin, 50 μg ml$^{-1}$ streptomycin, 1 mM sodium pyruvate, 10 μM HEPES, 55 μM 2-mercaptoethanol) supplemented with 1 ng ml$^{-1}$ hIL-2 (Peprotech). CTLL-2 proliferation was measured by incorporation of radiolabelled thymidine. Briefly, 5 × 10$^3$ cells were cultured in the presence of IL-2-Fc for 24 h, followed by addition of $^3$H thymidine and culturing for a further 24 h. Cells were collected onto a glass fibre filter and radioactivity was quantified using a liquid scintillation counter (Perkin Elmer). Prior to binding and signalling assays, CTLL-2 cells were starved of IL-2 for 6 h in order to remove surface-bound cytokine. Binding of IL-2-Fc variants to CTLL-2 cells (2 × 10$^5$ cells per well) was allowed for 20 min on ice. Unbound IL-2-Fc was removed by washing in flow cytometry buffer (2% FBS, 2 mM EDTA in PBS) and surface-bound IL-2-Fc was detected by flow cytometry using a FITC-conjugated anti-mIgG2a antibody (clone R19-15, BD). Signalling in CTLL-2 cells was measured by flow cytometric detection of pSTAT5 after IL-2-Fc stimulation. Cells (2 × 10$^5$ per well) were stimulated with IL-2-Fc variants for 10 min at 37 °C, followed by fixation (10 min at 37 °C in BD Cytofix), permeabilization (30 min on ice in BD Phosflow Perm Buffer III) and staining with an AF488-conjugated anti-pSTAT5 antibody (clone 47/Stat5-pY694, BD) for 45 min at RT.

***Ex vivo* stimulation of mouse splenocytes and human PBMC.** Red blood cell-depleted splenocytes from an IL-7 transgenic mouse were incubated with Fc-Block (anti-mouse CD16/CD32, BD) for 20 min on ice, followed by washing in flow cytometry buffer and staining with fluorophore-conjugated anti-CD8a and anti-CD44 antibodies. Stained cells were seeded in a 96-well plate (1 × 10$^6$ cells per well) and re-suspended in serially diluted IL-2-Fc variants. Stimulation was allowed for 10 min at 37 °C, followed by washing with flow cytometry buffer, fixation, permeabilization and intracellular staining with AF488-conjugated anti-pSTAT5 (clone 47/Stat5-pY694, BD). The levels of pSTAT5 in the CD8$^+$ CD44$^{high}$ population were determined by flow cytometry. For flow cytometry of human PBMC, the following antibodies were used: Pacific Blue-conjugated anti-CD8 (clone RPA-T8, BD), APC-conjugated anti-CD4 (clone S3.5, Thermo Fisher), PE-conjugated anti-FoxP3 (clone 259D/C7, BD). Stimulation of PBMC was allowed for 10 min at 37 °C in the presence of 1 μg ml$^{-1}$ (Treg stain) or 10 μg ml$^{-1}$ (CD8$^+$ T-cell stain) IL-2-Fc. Detection of intracellular pSTAT5 was performed as described above. Buffy coats from normal donors were obtained from the Australian Red Cross Blood Service. Informed consent was obtained from all subjects and approval for this study was obtained from the human research ethics committee of the St Vincent's Hospital (Sydney, Australia).

**Serum ELISA.** Mice were injected with a single dose (i.p.) of 20 μg hIL-2, 3 μg hIL-2 + 15 μg IL-2-Fc variants. Blood samples were taken at specified time points via tail vein bleeding and serum was separated by centrifugation after resting for 2 h at RT. hIL-2 was detected using an anti-hIL-2 mAb (clone 5344.111, BD) for capture and an anti-hIL-2 biotinylated polyclonal antibody (BAF202, R&D systems) for detection. Detection of IL-2/mAb and IL-2-Fc variants was performed using an anti-mIgG2a mAb (clone R11-89, BD) for capture and BAF202 for detection. Blocking and dilutions were performed in 1% bovine serum albumin in PBS and individual standard curves were built to determine the serum levels of each variant.

**Adoptive transfer of MP CD8 cells.** The MP CD8 reporter assay was performed as previously described[7]. MP CD8 cells from IL-7 Tg mice were fluorescence-activated cell sorting-sorted and labelled with carboxyfluorescein succinimidyl ester (CFSE) for adoptive transfer into C57BL/6 recipients. After cytokine treatment, donor MP CD8 cells in the spleens of recipient mice were identified by flow cytometry as Thy-1.1$^+$ and their proliferation measured by CFSE dilution.

**Fluorescence microscopy.** Spleen tissue from mice treated with Alexa Fluor 647-labelled IL-2-Fc was collected 12 h post injection and frozen in optimal cutting temperature (OCT) embedding medium (Sakura). Tissue sections (5 μm) were cut using a Leica CM3050 S Cryostat, followed by fixation in acetone (7 min) and re-hydration in PBS. After blocking with Protein Block solution (Dako), sections were stained with anti-B220-biotin, PE-conjugated anti-FoxP3 and BV421-conjugated anti-CD4 (in Antibody Diluent solution, Dako), followed by washing in PBS + 1% Tween 20 and staining with AF488-conjugated streptavidin (Thermo Fisher). Sections were mounted with Fluoromount (Sigma) and analysed using a Leica DM5500 microscope.

**Binding to macrophages *ex vivo*.** Red blood cell-depleted splenocytes from C57BL/6 mice were stained with anti-CD11b-PE, anti-F4/80-FITC and anti-CD3e-PE-Cy7. Stained cells were aliquoted into 96-well plates (1 × 10$^6$ cells per well), washed in flow cytometry buffer and incubated with 10 μg ml$^{-1}$ IL-2-Fc. After

IL-2-Fc binding, cells were washed in flow cytometry buffer and stained with $2 \mu g\, ml^{-1}$ anti-human IL-2-biotin (clone 5344.111, BD). Finally, cells were stained with streptavidin-APC (eBioscience) prior to flow cytometric analysis.

**Complement deposition assay.** CTLL-2 cells were collected 2–3 days after passaging and surface-bound hIL-2 was stripped by washing in RPMI 1640, 2% FBS, pH 3 for 20 s. Since complement deposition is calcium-dependent, all steps were performed in EDTA-free buffers. Accordingly, washing and incubation steps were performed in high-glucose DMEM supplemented with 0.5% bovine serum albumin and 0.08% sodium azide. IL-2-stripped CTLL-2 cells were aliquoted into 96-well plates ($3.5 \times 10^5$ cells per well) and re-suspended in $42 \mu g\, ml^{-1}$ IL-2-Fc (500 nM). After IL-2-Fc binding, cells were washed in media and re-suspended in purified mouse complement (Cedarlane Laboratories) diluted 1:2. Complement deposition was allowed for 2 h on ice, followed by washing and staining with anti-mIgG2a-AF488 and anti-mC1q-biotin (clone RmC7H8, Cedarlane Laboratories). Cells were then washed and stained with streptavidin-APC (eBioscience). Finally, cells were re-suspended in HEPES-buffered saline supplemented with 2% FBS and 1 mM $CaCl_2$ for flow cytometric analysis.

**Statistical analysis.** Data are displayed as mean ± s.e.m., (*$P < 0.05$, **$P < 0.01$, ***$P < 0.001$, ****$P < 0.0001$). Statistical analyses included one-way and two-way analysis of variance with Bonferroni *post hoc* test for multiple comparisons, and two-tailed unpaired Student's *t*-tests in data sets comparing two groups. Survival was displayed using Kaplan–Meier plots and compared by the Gehan–Breslow–Wilcoxon test. Data were analysed using the Prism software (GraphPad).

**Data availability.** The data supporting the findings of this study are available within the article and its Supplementary Information files and from the corresponding authors on reasonable request.

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

## Acknowledgements

This work was supported by the National Health and Medical Research Council. We thank Professor Alexander Rudensky (Howard Hughes Medical Institute and Memorial Sloan-Kettering Cancer Center, New York) for supplying the *Foxp3*<sup>DTR/GFP</sup> mice.

## Author contributions

D.C. and J.S. conceived and supervised the study. R.V.-L. designed and performed most of the experiments. T.G.P., K.E.W., J.S. and D.C. contributed to experimental design. C.L., D.Z., J.J., P.S., E.K.D. and C.K. performed or contributed to specific experiments. R.V.-L., J.S. and D.C. wrote the manuscript.

## Additional information

**Competing interests:** The authors declare no competing financial interests.

