## [Peer Review File · Nature Communications]

Reviewers' comments:

Reviewer #2 (Remarks to the Author):

This paper describes new IL-2-Ig fusion proteins. Although a number of these have been reported in the past, an interesting twist in this paper is that when FcR binding of the Ig portion of the fusion is intact, it appears to cause macrophage-mediated depletion of responding Treg cells to promote tumor immunity. They also report on Ig fusion proteins where IL-2 was mutated to select responses toward NK and CD8 memory cells. These molecules have activity similar to some IL-2/anti-IL-2 agonist complexes with selectivity to these cells. All new molecules in this report represent longer lasting IL-2s that might have uses in the clinic.

1) My main concern is about data presentation. Graphs show % increases. It would be easier to follow what is happening in vivo if results were reported as cell numbers. It would be useful to show overall number of CD4, CD8, NK, and B cells (or at least non-T, non NK cells) and then what happens to the cells of interest, Tregs, NK, and CD8 MPs. For example in Fig. 2B the authors show 4-6 fold increased in % of NK, CD8 MP and Tregs by IL-2-2x-Fc. Yet in Fig. 2A they report up to 15-30-fold increase in NK and CD8 MPs? How were these latter numbers derived? This suggests that there may be an overall increase in lymphoid cellularity that goes beyond the cell populations of interest. In a related way, it was hard to follow the results shown in Fig. 4A. It seems it would be clearer if cell numbers were shown in the graph to the left to follow cell expansion data shown to the right.

2) The authors interpret the improved tumor response by IL2-WT-Fc by depletion of Tregs? However, they never show fewer Tregs in the tumor lesion.

3) Why are not the CD8 MPs and NK cell not depleted through FcR mediated processes, as proposed for Tregs?

4) What does AF647-labeled IL-2 refer to? Please define.

5) In the text of the Result on page 10, please define what mAb is used for the IL-2/anti-IL-2 complexes.

6) A number of other IL-2Ig fusion proteins have already been described. Why was the Treg depleting activity not seen? Some discussion about these other molecules might be added.

7) In Fig. 1A, "single" or "double" in the graph is confusing and should be eliminated. The nomenclature on the x-axis is clear enough.

8) For Fig 1G middle panel, is nM correct? It seems it should be pM? IL-2-dependent proliferation and pSTAT5 of CTLL cells should be closer in dose responses than reported.

Reviewer #3 (Remarks to the Author):

This is a very well written and presented study which explores the use of IL2 for modulating anti-tumour immune responses. It builds on a body of information relating to use of IL2 in cancer treatments and the ability of IL2 / anti-IL2 mAb complexes to reduce IL2-mediated toxicities whilst preserving beneficial immune-modulating capacity.

The basic premise of the study was to design IL2 based reagents to be used at non-toxic doses for promoting beneficial responses. The approaches are elegant and combine mutagenesis, protein engineering, studies of protein-protein interactions to rationally design potentially therapeutic reagents. The conclusions indicate that IL2-Fc has superior anti-tumour activity.

The mechanisms underpinning this activity remain to be validated. In this respect the study would be significantly improved by showing what the anti-tumour activity is in IL2-Fc treated mice. Presumably CD8+ T cells? In addition, the depleting effect of the IL2-Fc is assessed only in spleens - this should also be assessed in draining lymph nodes and tumours as ADCC activity may be different at each site.

Reviewer #4 (Remarks to the Author):

Summary:

In their manuscript "Potent antitumour activity of IL-2-Fc fusion proteins through Fc-mediated depletion of regulatory T-cells," Vazquez-Lombardi et al implement a rigorous characterization of fusions comprised of the IL-2 cytokine and an antibody Fc domain in an effort to develop superior cancer immunotherapeutics and gain insight into their mechanisms for tumor clearance. The authors develop new IL-2-Fc fusions with ablated binding to the IL-2R α subunit in an effort to extend in vivo half-life of the IL-2 cytokine and selectively direct its signaling toward cytotoxic T lymphocytes and NK cells rather than TRegs to achieve enhanced anti-tumor activity. Unexpectedly, they find that their IL-2R α binding-deficient mutant IL-2 Fc fusions actually have reduced efficacy in tumor clearance compared to the wild type IL-2-Fc fusion due to their failure to engage TRegs and deplete them through antibody-dependent cellular cytotoxicity (ADCC). Their findings highlight the importance of Fc effector function considerations in immunotherapeutic design and establish that TReg depletion is critical for IL-2-mediated inhibition of tumor growth.

This mechanistic advance will be of interest for the field of immunotherapeutics as numerous Fc-fused drugs are currently in development. However, the authors' objective of engineering an enhanced IL-2-based cancer therapeutic has not been attained. The only construct that controls tumor growth in the in vivo models they present is the wild type IL-2-Fc fusion, which is not novel or unique to their study. Use of IL-2-Fc fusions to achieve improved disease therapy has been reported going back more than 15 years for a wide range of applications (for instance Zheng et al, J Immunol, 1999; Kang et al, Am J Transplant, 2007; Millington et al, J Heart Lung Transplant, 2012; Tzeng et al, PNAS, 2014; Mitra et al, Immunity, 2015).

Thus, the authors need to reshape their manuscript to emphasize the novelty of their mechanistic findings rather than the constructs they developed. Currently the manuscript reads like a hodge-podge of results, and too much emphasis is put on the mutational analysis of CD25 binding by IL-2. It is rather puzzling why the authors spent so much effort on this when there is an abundant literature (e.g. Thanos et al.) on mutational analysis of IL-2/CD25 binding. As such, this aspect of the study is far less interesting and important than the mechanistic finding about Treg depletion (although as the authors cite, the role of anti-tumor effect of Treg depletion by Fc-fusions has been demonstrated before, just not with IL-2-Fc fusions).

Collectively, there is a lot of interesting data in here, but the paper needs to be focused.

Specific Comments:

*The title of the manuscript should be changed to emphasize the novel contribution of their work. We suggest, "IL-2-Fc fusion anti-tumor activity requires Fc effector function-mediated depletion of regulatory T-cells."

*The concept of linking Fc domain to IL-2 is not novel; Many other examples of fusing IL-2 to Fc exist (for instance Zheng et al, J Immunol, 1999; Kang et al, Am J Transplant, 2007; Millington et al, J Heart Lung Transplant, 2012; Tzeng et al, PNAS, 2014; Mitra et al, Immunity, 2015). A review of previous literature on IL-2/Fc fusions (both lytic and nonlytic) should be included in the final paragraph of the Introduction and a comparison of their fusion construct to others designed

previously should be included in the Discussion.

*The authors make no mention of the well-characterized human IL-2 F42A mutation (Weir et al, *Biochemistry*, 1988), which has been demonstrated to abrogate IL-2 interaction with IL-2R α . Why was this not considered or compared in the IL-2 mutant screen?

*In Figures 1 and S1, the authors show SPR binding sensograms for their IL-2-Fc fusions against human IL-2R α and human IL-2R β , but all subsequent in vitro and in vivo characterization is performed against the mouse receptors. The authors should present SPR binding data against mouse IL-2R α and IL-2R β and discuss any differences in the absolute or relative affinities of these chains between the two species.

*In Figure S1G-J, the maximum RU for mutant IL-2-Fc fusions is approximately half that of the wild type IL-2-Fc fusion. Given the molecular weights are the same, it is unclear why this would be observed and this raises questions about the integrity and functionality of their expressed mutant IL-2-Fc fusions. As the quality of their expressed proteins and the concentrations of these constructs are integral to the conclusions they draw from signaling and functional assays, the authors should demonstrate integrity of their fusions by presenting non-reducing and reducing SDS-PAGE analysis as well as size exclusion chromatograms for the R38D, E61R, 2X, and 3X mutants.

*The induction of pulmonary oedema by IL-2/mAb complex and its ineffective control of tumor growth conflict with earlier results Boyman (Krieg et al, *PNAS*, 2010; Levin et al, *Nature*, 2012) demonstrating therapeutic efficacy of these constructs without inducing vascular leak syndrome in the B16F10 tumor model. Is this a dosage effect? The authors should address this apparent discrepancy in the Discussion section.

*The authors base all of their conclusions regarding the therapeutic efficacy and mechanism of their IL-2-Fc fusions on results from a single tumor model (the B16F10 mouse melanoma model). To test whether their conclusions hold more generally, at least one additional syngeneic tumor model should be used.

Minor Points:

*In Figure 1D, an accurate KD cannot be calculated from the data presented. Either higher concentrations of the IL-2-Fc fusion should be assayed or the KD should be indicated as >2 μ M.

*In the final paragraph of the Conclusion, the authors claim that they have "generated IL-2-Fc fusion proteins that significantly exceed the activity and efficacy of IL-2/mAb complexes." However, they have not demonstrated any therapeutic or dose tolerance advantage for IL-23XFc or any of their other mutants over IL-2/mAb treatment and, in fact, it seems to perform worse than the cytokine-antibody complex. The only superior molecule they present is a wild type IL-2-Fc fusion, which has been previously reported. The claim should be modified accordingly.

Response to reviewers' comments

In order to address the issues and suggestions raised we have implemented extensive changes and additions to our manuscript. More specifically, we now:

- Have significantly rewritten the manuscript in order to shift focus towards the characterization of IL-2^{WT}Fc antitumour activity. We have considerably shortened the initial result section relating to the generation of IL-2 mutants. Nevertheless, the generation of new mutants was essential for this study, and we demonstrate that previously described mutants do not sufficiently abolish binding to CD25 in the bivalent Fc-fusion format utilized here. These advances will be of interest to readers with a background in protein engineering and structure, and are now described in an expanded supplementary section.
- Provide detailed mechanistic insights into the enhanced antitumour activity displayed by IL-2^{WT}Fc (as requested by reviewers 2 and 3).
- Provide data relating to an additional *in vivo* model (CT26 colorectal carcinoma), which are in excellent agreement with the B16F10 melanoma data outlined in our initial submission.

Please find our detailed response as below.

Reviewer 2

This paper describes new IL-2-Ig fusion proteins. Although a number of these have been reported in the past, an interesting twist in this paper is that when FcR binding of the Ig portion of the fusion is intact, it appears to cause macrophage-mediated depletion of responding Treg cells to promote tumor immunity. They also report on Ig fusion proteins where IL-2 was mutated to select responses toward NK and CD8 memory cells. These molecules have activity similar to some IL-2/anti-IL-2 agonist complexes with selectively to these cells. All new molecules in this report represent longer lasting IL-2s that might have uses in the clinic.

We thank the reviewer for highlighting the novelty of our findings (“an interesting twist in this paper...depletion of responding Treg cells to promote tumour immunity”) and for providing suggestions and comments, which we address as below.

1) My main concern is about data presentation. Graphs show % increases. It would be easier to follow what is happening in vivo if results were reported as cell numbers. It would be useful to show overall number of CD4, CD8, NK, and B cells (or at least non-T, non NK cells) and then what happens to the cells of interest, Tregs, NK, and CD8 MPs. For example in Fig. 2B the authors show 4-6 fold increased in % of NK, CD8 MP and Tregs by IL-2-2x-Fc. Yet in Fig. 2A they report up to 15-30-fold increase in NK and CD8 MPs? How were these latter numbers derived? This suggests that there may be an overall increase in lymphoid cellularity that goes beyond the cell populations of interest. In a related way, it was hard to follow the results shown in Fig. 4A. It seems it would be clearer if cell numbers were shown in the graph to the left to follow cell expansion data shown to the right.

This is a valid comment and we have included several changes in the manuscript to address this issue. In particular, we have now included both cell numbers and percentages in Figure 3 (which represents a single experiment). However, other figures illustrate data from pooled experiments (Figures 1B, 1C and S3A in revised manuscript), and for those we show normalised data (i.e. fold expansion relative to PBS controls in each experiment) rather than absolute cell numbers. In order to clarify the apparent discrepancy between the expansion levels of MP CD8 and NK cells in terms of numbers (15-30 fold) and percentages (4-6-fold) after IL-2^{2X}Fc treatment, we now discuss this in detail in the Supplementary Discussion section (Figures S3A-S3F). Indeed, as mentioned by the reviewer, the superior expansion in MP CD8 and NK cell numbers is partially explained by a significant increase in spleen lymphoid cellularity (Figure S3B) but also by the previously mentioned increases in population frequencies (Figure S3C), a significant increase in the CD8 to CD4 T-cell ratio (Figure S3D) and a prolongation of serum half-life relative to the IL-2/mAb complexes and IL-2^{WT}Fc (Figures S3E and S3F).

Finally, we have comprehensively analysed the immune cellular biodistribution of fluorescently labelled IL-2-Fc proteins (Figures 4B and S5B). In addition to interaction of IL-2-Fc with T-cells, NK cells and NKT cells; these experiments also analysed interactions with B-cells, dendritic cells (DCs), macrophages and neutrophils. We found that association of IL-2-Fc variants with B-cells and DCs was generally low, while interaction with macrophages and neutrophils was substantially higher and FcγR-dependent (Figures S4B and S4C).

2) The authors interpret the improved tumor response by IL2-WT-Fc by depletion of Tregs? However, they never show fewer Tregs in the tumor lesion.

In order to gain further insights into the superior antitumour activity of IL-2^{WT}Fc therapy, we performed flow cytometric analysis of spleens, draining lymph nodes (dLNs) and B16F10 tumours in mice treated with IL-2^{WT}Fc (Treg-depleting), IL-2^{WT}Fc^{nil} (Treg-expanding) or PBS control (Figures 5B-5G and S9). Unlike what was observed in the spleen, Treg depletion in the tumour lesion was not evident (Figure 5B). However, tumours in the IL-2^{WT}Fc group displayed reduced frequencies of *activated* Tregs (Figure 5C), a large increase in the total influx of leukocytes (Figure 5D) and a significant enrichment of CD8⁺ T-cells (Figures 5E-5G). Our data thus suggests that the observed depletion of Tregs in the periphery (Figure 3) contributes to the establishment of tumours with increased immunogenicity and enhanced susceptibility to CD8-mediated antitumour immunity.

3) Why are not the CD8 MPs and NK cell not depleted through FcR mediated processes, as proposed for Tregs?

The selective depletion of Tregs results from high-affinity targeting of this subset by IL-2-Fc fusion proteins. While Tregs display the high-affinity IL-2R $\alpha\beta\gamma_c$, MP CD8 and NK cells express the lower affinity IL-2R $\beta\gamma_c$. This correlates with the substantially stronger binding of IL-2-Fc proteins to Tregs over CD8 or NK subsets, which we have demonstrated both *in vitro* (Figure 4A) and *in vivo* (Figures 4D-4G). Thus, the IL-2 component of IL-2-Fc fusion proteins is able to efficiently target high-affinity IL-2R $\alpha\beta\gamma_c$ on the surface of Tregs, while their Fc portion mediates engagement of Fc γ R on immune effector subsets, likely macrophages and neutrophils (Figure 4C), leading to selective Treg elimination through ADCC. Discussions relating to the affinity difference underpinning selective Treg depletion are presented in more detail on pages 9-11 of the revised manuscript.

4) What does AF647-labeled IL-2 refer to? Please define.

Alexa Fluor 647, used to label IL-2-Fc proteins in cellular biodistribution studies (Figures 4B-4G and Figure S5). In the interest of clarity, we have replaced 'AF647+' or 'AF647-labelled IL-2' for 'IL-2-Fc+' throughout the text and figures.

5) In the text of the Result on page 10, please define what mAb is used for the IL-2/anti-IL-2 complexes.

The utilized mAb is MAB602 (information now included in the last paragraph of page 5 in addition to Methods section).

6) A number of other IL-2Ig fusion proteins have already been described. Why was the Treg depleting activity not seen? Some discussion about these other molecules might be added.

This is a valid and highly relevant question. As described on page 9 of the revised manuscript, the Treg-depletive activity of IL-2^{WT}Fc only became evident after comparison to the effector-less IL-2^{WT}Fc^{nil} protein. While both proteins expanded MP CD8 and NK cell subsets equally, no Treg

expansion was observed for IL-2^{WT}Fc while substantial expansion of this subset was induced by IL-2^{WT}Fc^{nil} (Figures 3C and D). As discussed in the manuscript, these proteins were identical except for their ability to bind FcγR (Figure S7), thereby providing a strong indication of Treg-specific depletive activity by FcγR-binding constructs. Notably, Treg-depletion relative to PBS controls was nearly undetectable (Figures 3E and 3F). Thus, we speculate that previous reports (Jindal et al., 2015; Millington et al., 2012; Zheng et al., 2003; Zheng et al., 1999) have failed to identify Treg-depleting activity because a direct comparison between lytic and effector-less IL-2-Fc in the context of Treg expansion is required.

We should highlight a recent study that has described the development of IL-2-IgG molecules as fusions to the IgG light chain (Tzeng et al., 2015). Interestingly, comparison of lytic and effector-less versions of these molecules revealed no evidence of Treg-depletive activity. However, unlike our study, the cited study reports very low levels of macrophage binding *in vivo*, regardless of FcγR binding status. Thus, failure to identify Treg depletion could be possibly related to a reduced association with immune effector subsets responsible for ADCC, such as macrophages. This could in turn arise from steric hindrances exerted by the IL-2 component in their design (fused to the C-terminus of IgG light chain), which sits in close spatial proximity to the FcγR binding site.

7) In Fig. 1A, "single" or "double" in the graph is confusing and should be eliminated. The nomenclature on the x-axis is clear enough.

The words "single" and "double" have now been removed from this panel (Figure S3A in revised manuscript).

8) For Fig 1G middle panel, is nM correct? It seems it should be pM? IL-2-dependent proliferation and pSTAT5 of CTLL cells should be closer in dose responses than reported.

We should clarify that there is no typographic error and that the concentration of IL-2-Fc is indeed nanomolar for CTLL-2 proliferation data (Figure S2B, middle panel) [Note: Figure 1G in the original manuscript corresponds to figure S2B in the revised manuscript]. We agree with the reviewer that lower EC50 values are expected, given that the EC50 of monovalent IL-2 for CTLL-2 proliferation has been typically reported in the picomolar range. However, we should emphasize that many variables come into play in proliferation studies as cells are incubated for a total of 48 h at 37 °C in the presence of IL-2-Fc. For instance, there might be differences in the rates of receptor-mediated internalisation and degradation between our bivalent IL-2-Fc and monovalent unfused IL-2. For this reason, we also include STAT5 phosphorylation data (Figure S2B, right panel), in which cells are stimulated for only 10 min at 37 °C. Collectively, these assays aimed to assess relative differences in CTLL-2 activity amongst IL-2-Fc variants.

Reviewer 3

This is a very well written and presented study which explores the use of IL2 for modulating anti-tumour immune responses. It builds on a body of information relating to use of IL2 in cancer treatments and the ability of IL2 / anti-IL2 mAb complexes to reduce IL2-mediated toxicities whilst preserving beneficial immune-modulating capacity. The basic premise of the study was to design IL2 based reagents to be used at non-toxic doses for promoting beneficial responses. The approaches are elegant and combine mutagenesis, protein engineering, studies of protein-protein interactions to rationally design potentially therapeutic reagents. The conclusions indicate that IL2-Fc has superior anti-tumour activity.

The mechanisms underpinning this activity remain to be validated. In this respect the study would be significantly improved by showing what the anti-tumour activity is in IL2-Fc treated mice. Presumably CD8+ T cells? In addition, the depleting effect of the IL2-Fc is assessed only in spleens - this should also be assessed in draining lymph nodes and tumours as ADCC activity may be different at each site.

We thank the reviewer for highlighting the strengths of our manuscript (e.g. “very well written and presented study”...“the approaches are elegant”) and for suggesting ways of improving our study.

Following the suggestions, we performed a comparative flow cytometric analysis of spleens, draining lymph nodes and B16F10 tumours in mice receiving IL-2-Fc therapy (Figures 5B-5G and S9). More specifically, we compared the proportions and numbers of immune cell subsets in mice receiving IL-2^{WT}Fc (Treg-depleting), IL-2^{WT}Fc^{nil} (Treg-expanding) or PBS control. As described above in our response to reviewer 2 (question 2) and in pages 11-12 (Figures 5B-5G) of our revised manuscript, we observed Treg-depletive activity in the spleen (consistent with Figure 3) but no such effect was evident in the tumour lesion. Importantly, however, B16F10 tumours harvested from mice receiving IL-2^{WT}Fc treatment displayed reduced frequencies of *activated* Tregs (Figure 5C), a substantial and significant increase in total leukocyte infiltration (Figure 5D) and a significant enrichment of infiltrating CD8⁺ T-cells (Figures 5E-5G). Therefore, as predicted by the reviewer, our data suggest that antitumour immunity is mediated by CD8⁺ T-cells. We agree with the reviewer that this mechanistic insight significantly strengthens our study.

Reviewer 4

In their manuscript "Potent antitumour activity of IL-2-Fc fusion proteins through Fc-mediated depletion of regulatory T-cells," Vazquez-Lombardi et al implement a rigorous characterization of fusions comprised of the IL-2 cytokine and an antibody Fc domain in an effort to develop superior cancer immunotherapeutics and gain insight into their mechanisms for tumor clearance. The authors develop new IL-2-Fc fusions with ablated binding to the IL-2R α subunit in an effort to extend in vivo half-life of the IL-2 cytokine and selectively direct its signaling toward cytotoxic T lymphocytes and NK cells rather than TRegs to achieve enhanced anti-tumor activity. Unexpectedly, they find that their IL-2R α binding-deficient mutant IL-2 Fc fusions actually have reduced efficacy in tumor clearance compared to the wild type IL-2-Fc fusion due to their failure to engage TRegs and deplete them through antibody-dependent cellular cytotoxicity (ADCC). Their findings highlight the importance of Fc effector function considerations in immunotherapeutic design and establish that TReg depletion is critical for IL-2-mediated inhibition of tumor growth.

This mechanistic advance will be of interest for the field of immunotherapeutics as numerous Fc-fused drugs are currently in development. However, the authors' objective of engineering an enhanced IL-2-based cancer therapeutic has not been attained. The only construct that controls tumor growth in the *in vivo* models they present is the wild type IL-2-Fc fusion, which is not novel or unique to their study. Use of IL-2-Fc fusions to achieve improved disease therapy has been reported going back more than 15 years for a wide range of applications (for instance Zheng et al, J Immunol, 1999; Kang et al, Am J Transplant, 2007; Millington et al, J Heart Lung Transplant, 2012; Tzeng et al, PNAS, 2014; Mitra et al, Immunity, 2015).

Thus, the authors need to reshape their manuscript to emphasize the novelty of their mechanistic findings rather than the constructs they developed. Currently the manuscript reads like a hodge-podge of results, and too much emphasis is put on the mutational analysis of CD25 binding by IL-2. It is rather puzzling why the authors spent so much effort on this when there is an abundant literature (e.g. Thanos et al.) on mutational analysis of IL-2/CD25 binding. As such, this aspect of the study is far less interesting and important than the mechanistic finding about Treg depletion (although as the authors cite, the role of anti-tumor effect of Treg depletion by Fc-fusions has been demonstrated before, just not with IL-2-Fc fusions. Collectively, there is a lot of interesting data in here, but the paper needs to be focused.

We thank the reviewer for highlighting the relevance of our study (i.e. “This mechanistic advance will be of interest for the field of immunotherapeutics as numerous Fc-fused drugs are currently in development”) and for providing valuable feedback.

As suggested, we have now focused the manuscript towards the characterization of the unexpectedly potent antitumour activity of IL-2^{WT}Fc. In doing so, we have considerably reduced the first section of the manuscript describing the engineering of IL-2-Fc mutants with abolished CD25 binding. Accordingly, data related to the engineering process have now been included as supplementary information (Figures S1-S5), accompanied by a brief supplementary discussion section describing our step-wise mutagenesis approach.

We should emphasize, however, that complete abolition of CD25 interactions in bivalent IL-2-Fc molecules was not trivial. Elimination of CD25 binding *in vitro* required the introduction of at least two charge-reversal mutations (R38D and E61R), as measured by SPR (Figure S1) and CTLL-2 assays (Figure S2). However, this double mutant (IL-2^{2X}Fc) still mediated significant expansion of CD25⁺ Tregs *in vivo* (5-fold, Figures S3A and S3C). Abolition of Treg expansion *in vivo* required the introduction of a third mutation at position K43. To achieve this, we had to carefully screen for a suitable substitution that maintained the structural balance of IL-2-Fc, as some of the tested mutations at position K43 resulted in a complete loss of activity (Figure S4A).

Thus, while previous reports of unfused monovalent IL-2 containing single mutations have elegantly defined residues important for IL-2/CD25 binding *in vitro* (Thanos et al., 2006), we show that complete prevention of CD25⁺ cell activation *in vivo* using the bivalent IL-2-Fc format requires the introduction of multiple substitutions that need to be combined in a highly specific manner.

Specific Comments:

1) The title of the manuscript should be changed to emphasize the novel contribution of their work. We suggest, "IL-2-Fc fusion anti-tumor activity requires Fc effector function-mediated depletion of regulatory T-cells."

We appreciate this suggestion by the reviewer, which aims to emphasize the novelty of our study. Our manuscript is now entitled: "Potent antitumour activity of IL-2-Fc fusion proteins requires Fc-mediated depletion of regulatory T-cells"

2) The concept of linking Fc domain to IL-2 is not novel; Many other examples of fusing IL-2 to Fc exist (for instance Zheng et al, J Immunol, 1999; Kang et al, Am J Transplant, 2007; Millington et al, J Heart Lung Transplant, 2012; Tzeng et al, PNAS, 2014; Mitra et al, Immunity, 2015). A review of previous literature on IL-2/Fc fusions (both lytic and nonlytic) should be included in the final paragraph of the Introduction and a comparison of their fusion construct to others designed previously should be included in the Discussion.

As suggested, a review of previous IL-2-Fc literature has now been included in the final paragraph of the Introduction. We also now mention in the Discussion section (page 16) that these studies have not reported any evidence of Treg depletion after treatment with their IL-2-Fc or IL-2-IgG constructs. A discussion on why that might be the case can be found above in our extensive response to reviewer 2 (question 6).

3) The authors make no mention of the well-characterized human IL-2 F42A mutation (Weir et al, Biochemistry, 1988), which has been demonstrated to abrogate IL-2 interaction with IL-2R α . Why was this not considered or compared in the IL-2 mutant screen?

As suggested, we have now included SPR (Figure S1E) and *in vivo* data (Figure S3A) using recombinant IL-2^{F42A}Fc. We found that, in the IL-2-Fc format, the F42A substitution is not sufficient to abrogate binding to recombinant human CD25 (Figure S1E). This is in agreement with previous studies demonstrating that while the F42A substitution reduces CD25 interactions, residual binding and induction of proliferation of CD25⁺ cells is still detectable (Weir et al., 1988). Furthermore, the *in vivo* activity of F42A was substantially lower than the double and triple mutants developed here (Figures S3A and S4E).

4) In Figures 1 and S1, the authors show SPR binding sensograms for their IL-2-Fc fusions against human IL-2R α and human IL-2R β , but all subsequent *in vitro* and *in vivo* characterization is performed against the mouse receptors. The authors should present SPR binding data against mouse IL-2R α and IL-2R β and discuss any differences in the absolute or relative affinities of these chains between the two species.

We have now included SPR data showing binding of IL-2-Fc variants to the mouse IL-2R α subunit (mCD25, Figure S1G). Remarkably, mCD25 data showed nearly identical affinities and similar disruptions induced by mutation as experiments using human CD25 (Figures S1D and S1E). Assessment of binding to the IL-2R β (mCD122) subunit is not reported, as this chain is not commercially available and its recombinant production in-house was problematic. Indeed, the ectodomain of mCD122 (Ala 27- Glu 240) contains a free-cysteine that likely results in the observed

high levels of protein aggregation and poor levels of binding by both IL-2-Fc and anti-mCD122 positive control (data not shown). Notably, mutation of free-cysteines and fusion of mCD122 to the Fc region did not improve binding profiles. We have extensive data on mCD122 recombinant production and optimisation, which we can make available upon request.

5) In Figure S1G-J, the maximum RU for mutant IL-2-Fc fusions is approximately half that of the wild type IL-2-Fc fusion. Given the molecular weights are the same, it is unclear why this would be observed and this raises questions about the integrity and functionality of their expressed mutant IL-2-Fc fusions. As the quality of their expressed proteins and the concentrations of these constructs are integral to the conclusions they draw from signaling and functional assays, the authors should demonstrate integrity of their fusions by presenting non-reducing and reducing SDS-PAGE analysis as well as size exclusion chromatograms for the R38D, E61R, 2X, and 3X mutants.

In this comment, the reviewer refers to SPR data showing binding of IL-2-Fc variants to human CD122 (now Figure S1F in the revised manuscript). To clarify this issue we now provide SDS-PAGE and size-exclusion chromatography (SEC) profiles of expressed proteins (Reviewer Figs. 1 and 2, appended below). These data show that some single mutations result in increased levels of protein aggregation (IL-2^{E61R}Fc), while others actually improve protein aggregation profiles relative to the parental IL-2^{WT}Fc protein (IL-2^{R38D}Fc). Importantly, both the IL-2^{2X}Fc double and IL-2^{3X}Fc triple mutants display lower levels of protein aggregation than IL-2^{WT}Fc, which correlate to their higher yields of recombinant protein expression (Reviewer Fig. 3). Thus, our data suggests that protein integrity does not play a role in the observed reductions in RU amplitude of CD122 binding. For instance, the IL-2^{R38D}Fc single mutant shows enhanced protein integrity (Reviewer Fig. 2) but nevertheless reduced RU levels (Figure S1F, second column). Thus, a likely explanation for reduced RU amplitude is that the introduced CD25-disrupting mutations have a minimal collateral impact on CD122 binding.

6) The induction of pulmonary oedema by IL-2/mAb complex and its ineffective control of tumor growth conflict with earlier results Boyman (Krieg et al, PNAS, 2010; Levin et al, Nature, 2012) demonstrating therapeutic efficacy of these constructs without inducing vascular leak syndrome in the B16F10 tumor model. Is this a dosage effect? The authors should address this apparent discrepancy in the Discussion section.

As highlighted by the reviewer, the cited studies demonstrated that low-dose IL-2/mAb therapy was less toxic and more efficacious than high-dose (HD) treatment with uncomplexed human IL-2. However, we should mention that although toxicity was significantly reduced relative to HD IL-2, IL-2/mAb treatment still induced residual toxicity, as evidenced by ~ 50% increases in pulmonary wet weight (Krieg et al., 2010; Levin et al., 2012). This is in excellent agreement with our data (Figure 2B). Furthermore, while low-dose IL-2/mAb resulted in significantly delayed B16F10 tumour growth compared to HD IL-2 and PBS controls, no tumour rejection was observed and maximal tumour loads were eventually reached (Krieg et al., 2010). This is similar to what we observed in our experiments (Figure 2F). Note: The discussion above compares identical dosages of IL-2/mAb. In Krieg *et al.* 2010, this corresponds to five i.p. IL-2/mAb injections containing 15,000 Units of IL-2 each, which is equivalent to the 1 µg IL-2 + 5 µg mAb per dose utilized in Levin et al. 2012 and in our study.

6) The authors base all of their conclusions regarding the therapeutic efficacy and mechanism of their IL-2-Fc fusions on results from a single tumor model (the B16F10 mouse melanoma model). To test whether their conclusions hold more generally, at least one additional syngeneic tumor model should be used.

Following the reviewer's suggestion, we tested the activity of IL-2-Fc fusion proteins and IL-2/mAb complexes in the CT26 colorectal carcinoma tumour model (page 13 and Figure 6B). Our results are highly consistent with the previously tested B16F10 melanoma model (Figures 5 and 6A), highlighting the Treg-depleting IL-2^{WT}Fc protein as the most efficacious therapeutic.

Minor Points:

In Figure 1D, an accurate KD cannot be calculated from the data presented. Either higher concentrations of the IL-2-Fc fusion should be assayed or the KD should be indicated as >2 μ M.

The suggested change has now been incorporated into the relevant figure (Figure S1D in the revised manuscript).

In the final paragraph of the conclusion, the authors claim that they have "generated IL-2-Fc fusion proteins that significantly exceed the activity and efficacy of IL-2/mAb complexes." However, they have not demonstrated any therapeutic or dose tolerance advantage for IL-21XFc or any of their other mutants over IL-2/mAb treatment and, in fact, it seems to perform worse than the cytokine-antibody complex. The only superior molecule they present is a wild type IL-2-Fc fusion, which has been previously reported. The claim should be modified accordingly

We apologize for the confusion. The engineered IL-2^{3X}Fc mutant showed higher activity than IL-2/mAb complexes, as demonstrated by increased lymphocyte expansion profiles and selectivity (Figures 1B and 1C); while the IL-2^{WT}Fc protein displayed higher antitumour efficacy (Figures 5 and 6). We have modified the conclusion to remove any ambiguity (page 17): "In summary, here we describe the development and characterisation of IL-2-Fc fusion proteins that significantly exceed the specificity (IL-2^{3X}Fc) and efficacy (IL-2^{WT}Fc) of IL-2/mAb complexes."

Reviewer Fig. 1. SDS-PAGE analysis of IL-2-Fc fusion proteins. IL-2-Fc variants were produced in HEK-293 cells and purified using Protein G agarose beads. Expressed proteins were then filtered through a 0.22 μ m syringe-driven filter and analysed by SDS-PAGE under non-reducing (left) and reducing (right) conditions.

Reviewer Fig. 2. Size-exclusion chromatography (SEC) profiles of IL-2-Fc fusion proteins. Recombinant IL-2-Fc variants were filtered through a 0.22 μm syringe driven filter prior to SEC analysis. 100 μg of protein in PBS were then loaded onto a Superdex 75 10/300 GL column connected to an AKTA instrument for analysis.

Reviewer Fig. 3. IL-2-Fc recombinant expression in HEK293 cells. The engineered IL-2^{2X}Fc and IL-2^{3X}Fc fusion proteins display significantly higher expression yields relative to the parental IL-2^{WT}Fc.

References

- Jindal, R., Unadkat, J., Zhang, W., Zhang, D., Ng, T., Wang, Y., Jiang, J., Lakkis, F., Rubin, P., and Lee, W. (2015). Spontaneous Resolution of Acute Rejection and Tolerance Induction With IL-2 Fusion Protein in Vascularized Composite Allotransplantation. *Am. J. Transplant.* *15*, 1231-1240.
- Krieg, C., Letourneau, S., Pantaleo, G., and Boyman, O. (2010). Improved IL-2 immunotherapy by selective stimulation of IL-2 receptors on lymphocytes and endothelial cells. *Proc. Natl. Acad. Sci. U. S. A.* *107*, 11906-11911.
- Levin, A.M., Bates, D.L., Ring, A.M., Krieg, C., Lin, J.T., Su, L., Moraga, I., Raeber, M.E., Bowman, G.R., Novick, P., *et al.* (2012). Exploiting a natural conformational switch to engineer an interleukin-2 'superkine'. *Nature* *484*, 529-533.
- Millington, T., Koulmanda, M., Ng, C., Boskovic, S., Nadazdin, O.M., Benichou, G., Zheng, X.X., Strom, T.B., and Madsen, J.C. (2012). Effects of an agonist interleukin-2/Fc fusion protein, a mutant antagonist interleukin-15/Fc fusion protein, and sirolimus on cardiac allograft survival in non-human primates. *J. Heart Lung Transplant.* *31*, 427-435.
- Thanos, C.D., DeLano, W.L., and Wells, J.A. (2006). Hot-spot mimicry of a cytokine receptor by a small molecule. *Proc. Natl. Acad. Sci. U. S. A.* *103*, 15422-15427.
- Tzeng, A., Kwan, B.H., Opel, C.F., Navaratna, T., and Wittrup, K.D. (2015). Antigen specificity can be irrelevant to immunocytokine efficacy and biodistribution. *Proceedings of the National Academy of Sciences* *112*, 3320-3325.
- Weir, M.P., Chaplin, M.A., Wallace, D.M., Dykes, C.W., and Hobden, A.N. (1988). Structure activity relationships of recombinant human interleukin-2. *Biochemistry (Mosc.)* *27*, 6883-6892.

Zheng, X.X., Sanchez-Fueyo, A., Sho, M., Domenig, C., Sayegh, M.H., and Strom, T.B. (2003). Favorably tipping the balance between cytopathic and regulatory T cells to create transplantation tolerance. *Immunity* *19*, 503-514.

Zheng, X.X., Steele, A.W., Hancock, W.W., Kawamoto, K., Li, X.C., Nickerson, P.W., Li, Y.S., Tian, Y., and Strom, T.B. (1999). IL-2 receptor-targeted cytolytic IL-2/Fc fusion protein treatment blocks diabetogenic autoimmunity in nonobese diabetic mice. *J. Immunol.* *163*, 4041-4048.

Reviewers' comments:

Reviewer #2 (Remarks to the Author):

No new comments. The authors have satisfactorily addressed my concerns.

Reviewer #3 (Remarks to the Author):

I raised two key points previously:

1. the authors should demonstrate whether Treg depletion was observed in lymph nodes and tumour as well as in spleen

Figure 5 appears to show no alteration in Treg proportions in either compartment upon administration of IL2-WTFC. This seems at odds to me to the data in Figure 3 and adds nothing to clear up how efficient depletion is or isn't in the different tissues. The best data is when T cell numbers are expressed per gram of tissue but Tregs are not included in this analysis.

2. Demonstrate that CD8+ T cells are important for tumour rejection.

The simple experiment has not been done as far as I can see although the authors are happy to infer that CD8s are important as they are increased in treated tumours. I'm surprised that the experiment is not done given that the authors actually agree that this "mechanistic insight (which is inferred) significantly strengthens our study"!

Reviewer #4 (Remarks to the Author):

The revised manuscript is acceptable for publication

Response to reviewers' comments

We thank the editor and reviewers for the detailed comments. We have now addressed the concerns raised by Reviewer #3 by providing additional experimental data that:

- Demonstrate Treg depletion in dLN/spleen/tumour tissue after IL-2^{WT}Fc treatment (Figures S9A and S9B)
- Demonstrate that IL-2^{WT}Fc efficacy is critically dependent on CD8⁺ T-cells (Figures S9D and S9E)

Please find our detailed response as below.

Reviewer #2 (Remarks to the Author):

No new comments. The authors have satisfactorily addressed my concerns.

Reviewer #4 (Remarks to the Author):

The revised manuscript is acceptable for publication.

Reviewer #3 (Remarks to the Author):

1. the authors should demonstrate whether Treg depletion was observed in lymph nodes and tumour as well as in spleen Figure 5 appears to show no alteration in Treg proportions in either compartment upon administration of IL2-WTFC. This seems at odds to me to the data in Figure 3 and adds nothing to clear up how efficient depletion is or isn't in the different tissues.

As mentioned by the reviewer, we observed no evidence of Treg depletion in the tumour lesion in our original experiment (Figure 5B), in which flow cytometric analysis of tumours was performed 48 hours after the final dose of IL-2-Fc. In order to address this apparent discrepancy with our previous data showing Treg depletion in the spleen (Figure 3), we performed additional

experiments in which flow cytometric analysis of tissues was performed 24 hours after the last dose of IL-2-Fc (exactly as in Figure 3). At this time point we identified significantly reduced Treg proportions (Figure S9A) and total numbers (Figure S9B) in the spleen, dLN and tumours of IL-2^{WT}Fc-treated mice relative to their IL-2^{WT}Fc^{nil}-treated counterparts. Our new data thereby provides evidence for IL-2^{WT}Fc-mediated Treg-depletion for all analyzed tissues (see page 12, line 8).

2. Demonstrate that CD8+ T cells are important for tumour rejection. The simple experiment has not been done as far as I can see although the authors are happy to infer that CD8s are important as they are increased in treated tumours. I'm surprised that the experiment is not done given that the authors actually agree that this "mechanistic insight (which is inferred) significantly strengthens our study"

In order to address the reviewer's concern we performed additional B16F10 tumour growth experiments in which IL-2^{WT}Fc treatment was administered to mice receiving either CD8-depleting (Figure S9D) or CD4-depleting antibodies (Figure S9E). In excellent agreement with our flow cytometry data supporting a crucial role of CD8⁺ T-cells in the antitumour activity of IL-2^{WT}Fc (Figures S5E-G), depletion of this subset severely and significantly impaired the efficacy of IL-2^{WT}Fc treatment (Figure S9D). Conversely, depletion of CD4⁺ T-cells improved the efficacy of IL-2^{WT}Fc treatment (Figure S9E), presumably due to further depletion of CD4⁺ CD25⁺ Tregs (see page 12, line 23).

REVIEWERS' COMMENTS:

Reviewer #3 (Remarks to the Author):

Thank you very much. My concerns are addressed.